# Biased cytochrome P450-mediated metabolism via small-molecule ligands binding P450 oxidoreductase

Simon Bo Jensen [1], Sara Thodberg [2,3,4], Shaheena Parween [5,6], Matias E. Moses [1], Cecilie C. Hansen[2,3,4], Johannes Thomsen [1], Magnus B. Sletfjerding [1], Camilla Knudsen [2,3,4], Rita Del Giudice [2,3,4], Philip M. Lund [1], Patricia R. Castaño[5,6], Yanet G. Bustamante [1], Maria Natalia Rojas Velazquez[5,6], Flemming Steen Jørgensen [7], Amit V. Pandey [5,6], Tomas Laursen [2,3,4], Birger Lindberg Møller [2,3,4,8] & Nikos S. Hatzakis [1,9 ✉]

Metabolic control is mediated by the dynamic assemblies and function of multiple redox enzymes. A key element in these assemblies, the P450 oxidoreductase (POR), donates electrons and selectively activates numerous (>50 in humans and >300 in plants) cytochromes P450 (CYPs) controlling metabolism of drugs, steroids and xenobiotics in humans and natural product biosynthesis in plants. The mechanisms underlying POR-mediated CYP metabolism remain poorly understood and to date no ligand binding has been described to regulate the specificity of POR. Here, using a combination of computational modeling and functional assays, we identify ligands that dock on POR and bias its specificity towards CYP redox partners, across mammal and plant kingdom. Single molecule FRET studies reveal ligand binding to alter POR conformational sampling, which results in biased activation of metabolic cascades in whole cell assays. We propose the model of biased metabolism, a mechanism akin to biased signaling of GPCRs, where ligand binding on POR stabilizes different conformational states that are linked to distinct metabolic outcomes. Biased metabolism may allow designing pathway-specific therapeutics or personalized food suppressing undesired, disease-related, metabolic pathways.

[1] Department of Chemistry & Nanoscience Centre, University of Copenhagen, Copenhagen Ø, Denmark. [2] Plant Biochemistry Laboratory, Department of Plant and Environmental Sciences, University of Copenhagen, Copenhagen, Denmark. [3] Center for Synthetic Biology, Copenhagen, Denmark. [4] VILLUM Research Center "Plant Plasticity", Copenhagen, Denmark. [5] Pediatric Endocrinology, Diabetology, and Metabolism, University Children's Hospital, Bern, Switzerland. [6] Department of Biomedical Research, University of Bern, Bern, Switzerland. [7] Department of Drug Design and Pharmacology, University of Copenhagen, Copenhagen Ø, Denmark. [8] Carlsberg Research Laboratory, Copenhagen V, Denmark. [9] Novo Nordisk Foundation Centre for Protein Research, Faculty of Health and Medical Sciences, University of Copenhagen, Copenhagen N, Denmark. ✉email: hatzakis@chem.ku.dk

Dynamic assemblies and function of multiple redox protein partners with cytochromes P450 (CYPs) and NADPH-dependent cytochrome P450 oxidoreductase (POR) orchestrate control of multiple metabolic cascades across kingdoms[1–3]. POR transfers electrons to the heme iron of CYPs and other redox partners selectively activating them[4–10]. In plants, the coordinated assembly of POR-CYP complexes in dynamic metabolons enables on demand in vivo production of natural products to fend off, counteract or adapt to biotic or abiotic environmental stress, as recently demonstrated in the crop plant sorghum[1,11]. Metabolon formation also regulates parts of primary metabolism. In humans, POR-CYP assemblies serve to properly balance the metabolism of drugs, steroids, fatty acids, xenobiotics, and bio-active plant natural products in foods[2,4]. Mutations in human POR alter POR specificity towards activation of CYPs leading to severe disorders with multiple clinical manifestations varying from skeletal malformations with craniosynostosis (similar to Antley-Bixler Syndrome) to ambiguous genitalia and disorder of sexual development, amongst others[3,4,12–14]. Exploiting this regulatory layer is central for the treatment of metabolic disorder and tailored biosynthesis of natural products, however the mechanisms regulating, or biasing, POR specificity towards CYPs are not well understood.

Biased specificity has historically been observed to underlie function of signaling hubs like the G protein-coupled receptors (GPCRs). Docking of structurally diverse ligands biases GPCR conformational sampling, stabilizing distinct conformational states and thus the corresponding signaling pathways, a phenomenon called biased agonism[15–17]. POR acts as a metabolic hub: its conformational sampling and specificity towards CYPs is dependent on regulatory cues[5,6,18] and mutations[4,19–21] indicating that POR specificity of activating metabolic cascades may operate via mechanisms akin to biased agonism. However, to date no small molecules are known to target metabolic hubs like POR and allosterically control downstream metabolic pathways.

Here, we show that the specificity of POR towards diverse electron acceptors can be tuned by small molecules. We demonstrate using computational modeling that the small molecules serve as ligands binding on POR. Comparative in vitro activity assays on a set of diverse electron acceptors display that the ligands bias the specificity of both human and plant POR rather than inhibiting their function. Single molecule Förster (or Fluorescence detected) Resonance Energy Transfer (smFRET) provides mechanistic insights showing that ligand binding biases conformational sampling of plant POR, providing a link from biased conformational sampling to biased redox partner specificity. Lastly, we show that ligands alter CYP-mediated steroid hormone metabolism in human cells and microsomes emphasizing the biological relevance and applicability of controlling metabolic outcomes by targeting POR. Our data support a model of biased metabolism, a mechanism akin to biased signaling of GPCRs: POR conformational states are optimized to interact with certain CYPs and are linked to distinct downstream metabolic outcomes. Ligand-mediated control of POR conformational sampling thus appears to inhibit the activation of a subset of CYPs and/or enhance activation of others offering a new paradigm of metabolic control.

## Results

**Binding of POR ligands induce biased specificity towards electron acceptors.** Using computational docking simulations on POR crystal structures, we assessed the possible binding of small-molecule ligands that are known to affect the function of specific CYPs (Fig. 1A). Initial ligand selection was based on combination of the annotated subset of chemicals and approved drug subset from Drugbank following Lipinski Rule-of-Five descriptors and principal component analysis (see Supplementary material and Supplementary Fig. 1A). Based on docking simulations and functional activity assays, three structurally diverse ligands with promising effects on POR function were selected for detailed studies (Fig. 1B). Tested molecules showing weaker or no effects on POR function are displayed in Supplementary Fig. 1B. The three promising candidates were; (a) rifampicin, an antibiotic that induces expression and function of several CYPs[22], (b) cyclophosphamide, a chemotherapeutic prodrug which induces expression of several CYPs[23], and (c) dhurrin, a plant defense compound which biosynthesis requires the coordinated assembly and function of POR, two CYPs and a glucosyltransferase in dynamic metabolons[1,11].

Potential binding sites on POR were identified based on a SiteMap analysis[24] on human POR in a compact conformation (PDB 3QE2[13]) and using rat POR in an extended conformation (PDB 3ES9[7]). Rat POR shows 94% sequence identity with human POR (see Supplementary Fig. 2 for sequence alignment) and was chosen in the absence of a human POR extended conformation and in preference of the available human-yeast chimeric structure (PDB 3FJO[25]). The analysis on the two isoforms yielded five binding sites on each structure (see Supplementary Fig. 3A and Supplementary Table 1 for site scores, exposure and enclosure parameters). The ligands were docked into these binding sites, and all displayed a clear preference towards Site I on both structures with estimated binding energies ranging from −5 to −7 kcal/mol (see Supplementary Table 2). On human POR, Site I extends throughout the interface between the FMN-, FAD- and NADPH-binding domains comprising a relatively large volume. Both cyclophosphamide and dhurrin docked into two subsites of human POR Site I, called Ia and Ib, with almost equal binding energies, while rifampicin only docked into Site Ia (Fig. 1B and Supplementary Fig. 4). Site Ia forms a cavity partly comprised by the FAD and NADPH cofactors with the docked ligands extensively exposed to the surrounding solvent, while Site Ib lies at the interface between the FMN- and FAD-binding domains distant from the cofactors and further embedded into the protein structure. All three ligands were predicted to interact via H-bonds and π-π stacking to several amino acid residues on both POR isoforms (see Supplementary Figs. 3 and 4 and Supplementary Table 2). 1 microsecond molecular dynamics (MD) simulations confirmed these interactions and revealed stable binding for all three ligands upon an initial equilibration phase (see Supplementary Fig. 5 for detailed interactions and Supplementary Movies 1–4 displaying binding modes for each ligand). Tested ligands were found to interact primarily with amino acids that are conserved between human, rat and plant POR (see Supplementary Fig. 2 for sequence alignment), albeit each ligand interacted in different ways with additional specific amino acids (see Supplementary Fig. 5). Interestingly, all three ligands were found to interact directly with either one or both of amino acid residues G539 and R600 of human POR (see Supplementary Fig. 4 and Supplementary Table 2). Mutations of the very same residues are associated with POR deficiency. Patients with pathogenic mutations G539R and R600W suffer from disorder of sexual development due to low production of sex steroids indicating that this site is important for POR specificity towards steroid-metabolizing CYPs[4,12].

Our in vitro studies showed these ligands to bind on POR and regulate its function. Ligand binding on POR for all three ligands was confirmed by intrinsic fluorescence quenching assays (see Supplementary Methods and Supplementary Fig. 6). The recorded quenching was in agreement with MD simulations showing strong interaction of all three ligands with tryptophan and tyrosines (W679, Y458, Y481) in both site Ia and Ib

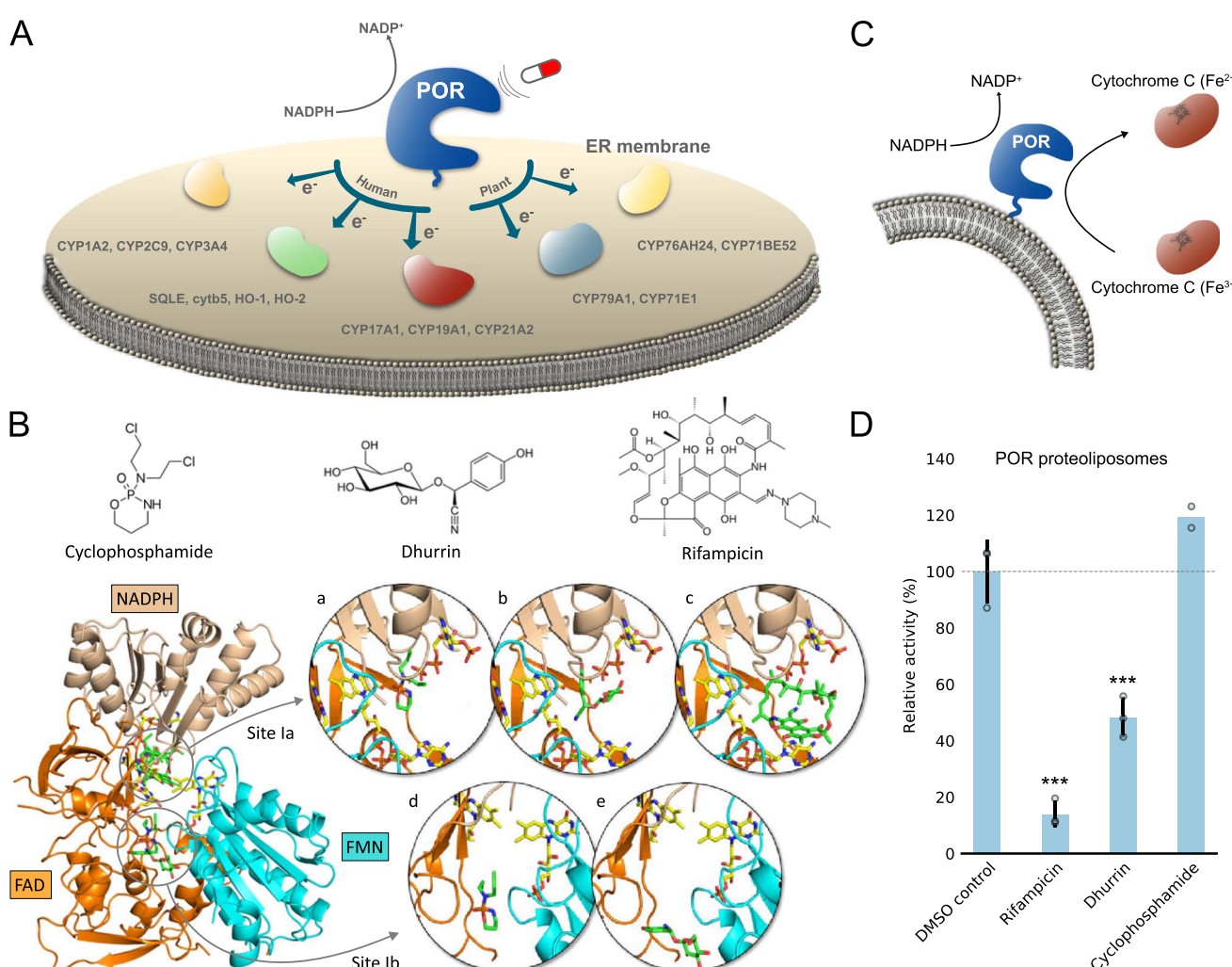

**Fig. 1 Small molecules dock on human POR and regulate electron transfer in vitro. A** POR is the omnipotent electron donor to all CYPs in the ER membrane, activating metabolic cascades in both human and plants by transferring electrons to redox partners. Targeting POR with small-molecule ligands may bias metabolic outcomes and regulate basic metabolism in humans or tune the formation of natural products in plants. **B** Molecular structures of small-molecule ligands and their respective binding. Ligands (green) were docked on human POR with cofactors (yellow) in a compact conformation (PDB 3QE2) in Sites Ia and Ib determined from SiteMap analysis (see Supplementary Fig. 3A and Supplementary Table 1). Insets display the predicted binding conformations of cyclophosphamide (a + d), dhurrin (b + e), and rifampicin (c). See Supplementary Figs. 3–5 for higher magnification and detailed interactions and Supplementary Table 2 for binding energies. **C** In vitro activity of human POR proteoliposomes measured by the commonly used Cyt$c$ assay[29]. **D** Ligands bias human POR capacity to reduce Cyt$c$ in proteoliposomes at 100 μM, acting either as agonist (cyclophosphamide) or inverse agonizts (dhurrin and rifampicin). The bar plot represents the mean ± SD of independent replicates ($n = 2$–3; see Supplementary Table 3 for exact value of $n$ for each experimental condition). Overlapping data points appear shaded. Level of significance is determined by one-way ANOVA and Tukey's HSD test correcting for multiple comparisons (*$p < 0.05$; **$p < 0.01$; ***$p < 0.005$; see Supplementary Methods and Supplementary Table 3 for details). Source data are provided as a Source Data file.

(Supplementary Fig. 5)[26]. To quantify the effect of ligand binding on POR function we performed in vitro functional assays on human POR proteoliposomes using cytochrome $c$ (Cyt$c$) as an electron acceptor (Fig. 1C, D, see Supplementary Methods and Supplementary Fig. 7 for spectral controls). All three ligands displayed a strong effect on the capacity of POR to reduce Cyt$c$. Rifampicin caused an activity decrease to 14 ± 5% of control, defined as POR activity without drug in otherwise identical conditions. Dhurrin caused a decrease to 48 ± 9% of control, while cyclophosphamide appeared to cause an increase to 119 ± 14% of control. All compounds were tested at 100 μM using 40 μM Cyt$c$ and 100 μM NADPH as substrates. The fact that the bound ligands modulate activity supports that POR can be a target for metabolic regulation and a modulator of therapeutic

activities. The observations that rifampicin and cyclophosphamide are known to interact with liver CYPs like CYP2B6[27] and CYP3A4[28] furthermore shows that direct binding on POR should be taken into consideration when screening for drug-CYP interactions.

Given that point-mutations in human POR can lead to altered specificity towards CYP isoforms[4,19], we tested whether small-molecule ligands can bias the specificity of human POR to reduce diverse electron acceptors. Comparative in vitro activity assays were carried out using commonly employed artificial electron acceptors of POR; Cyt$c$[29,30], resazurin (RS)[31] or 3-(4,5-dimethylthiazol-2-yl)-2,5-diphenyltetrazolium bromide (MTT)[32] (Fig. 2A). Each of the assays relies on spectral changes of the electron acceptor upon reduction by POR and thus directly

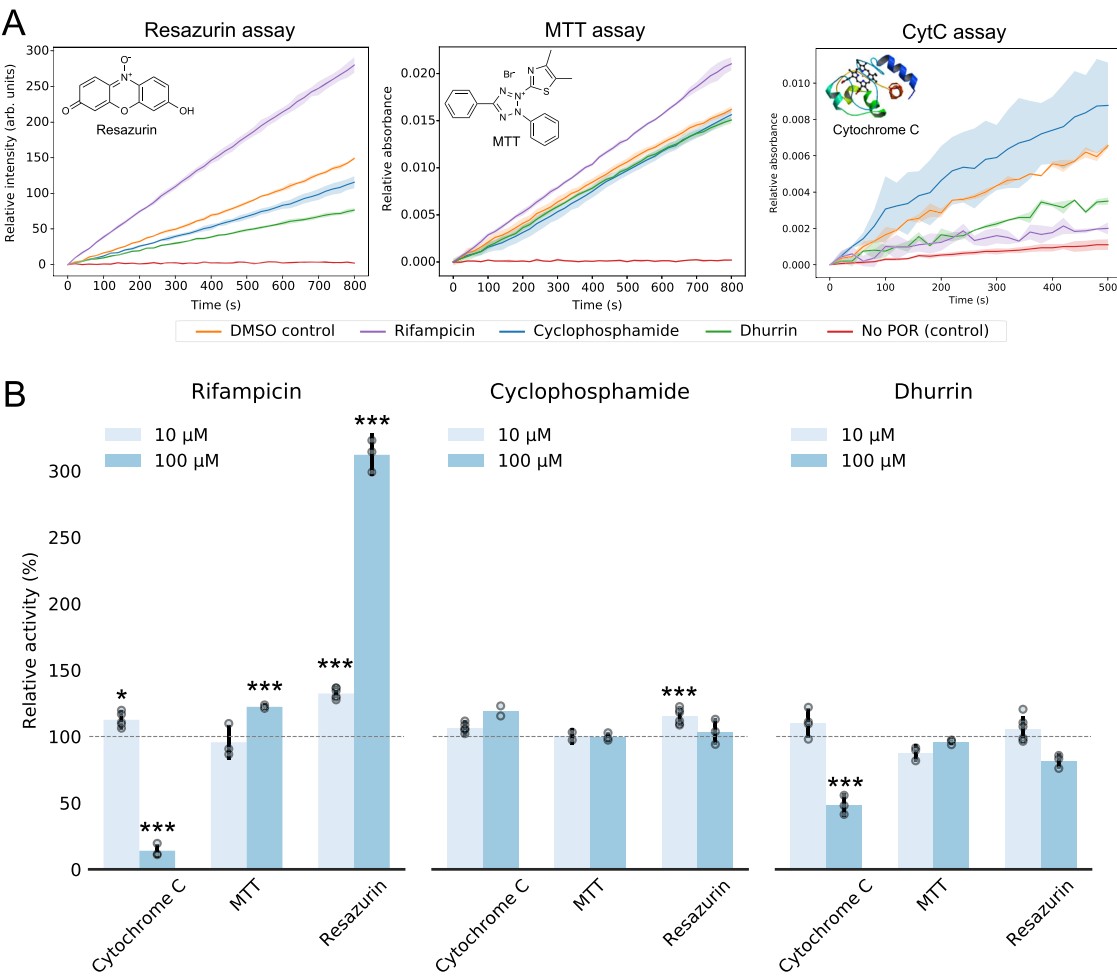

**Fig. 2 Small-molecule ligands bias specificity of human POR to reduce diverse electron acceptors. A** Human POR proteoliposome activity to reduce diverse electron acceptors was assessed using 100 µM NADPH and 10 µM RS (left), 500 µM MTT (middle) or 40 µM Cyt*c* (right) by monitoring changes in absorbance (550 nm for Cyt*c*, 610 nm for MTT) or fluorescence (582 nm for RS). Note the increased noise due to less sensitive UV–VIS readout for Cyt*c*. All activity traces depict the mean ± SD of at least three independent measurements. POR activity was extracted by fitting the linear region of the traces. **B** Ligands affect the electron donating capacity of human POR differentially dependent on the electron acceptor indicating biased specificity. Rifampicin reduces POR activity towards Cyt*c*, has a small effect on MTT reduction and enhances POR activity to reduce the electron acceptor resazurin by 3-fold. Cyclophosphamide results in minute increased activity towards Cyt*c*, while dhurrin reduces activity towards Cyt*c*. The bar plot represents the mean ± SD of independent replicates normalized to DMSO controls with propagated error ($n = 2$–6; see Supplementary Methods and Supplementary Table 3 for exact value of n for each experimental condition). Note, overlapping data points appear shaded. Data for Cyt*c* from Fig. 1 are included for comparison. All data are corrected for potential ligand photophysical effect (see Supplementary Fig. 7). Level of significance is determined by one-way ANOVA and Tukey's HSD test correcting for multiple comparisons (*$p < 0.05$; **$p < 0.01$; ***$p < 0.005$; see Supplementary Material for details). Source data are provided as a Source Data file.

reports on POR activity towards reducing the specific redox partner (see Supplementary Methods for details). Being comparative these assays exclude artefacts that may originate from potential FMN loss. Cyclophosphamide caused an increase in POR capacity to reduce resazurin (115 ± 7% of control at 10 µM) and appeared to cause an increase in POR capacity to reduce Cyt*c* (119 ± 14% of control at 100 µM) but had no significant effect in neither the MTT when tested at 10 or 100 µM (Fig. 2B). This supports that binding close to FAD and NADPH cofactors does not reduce or eliminate activity by non-specifically hindering POR motions and prohibiting electron transfer between FAD and FMN. Dhurrin, which caused an activity decrease in the Cyt*c* assay (48 ± 9% of control), only showed very minor effects in the MTT and RS assays (96 ± 3% and 82 ± 6% of control, respectively), see also Supplementary Fig. 8 for dose-response curves of dhurrin on hPOR in microsomes. The presence of rifampicin

caused a dramatic increase in POR capacity to reduce RS (312 ± 16% of control) and a smaller but significant increase in reduction of MTT (122 ± 3% of control; see Fig. 2B) (see Supplementary Table 3 for detailed data). This is striking as rifampicin decreased POR capacity to reduce Cyt*c* (14 ± 5% of control at 100 µM) and highlights that small-molecule ligands can bias the specificity of POR towards reducing diverse electron acceptors in a way similar to biased agonism of GPCRs.

**POR ligand binding and biased specificity pertain across mammal and plant kingdoms.** POR plays an omnipotent role as an electron donor to microsomal CYPs in all eukaryotes and serves as a key metabolic hub in plants as well as humans[1]. To test whether the effect of small-molecule ligand binding to POR is an omnipotent phenomenon underlying the regulation of POR in organisms from different kingdoms, we performed binding assays

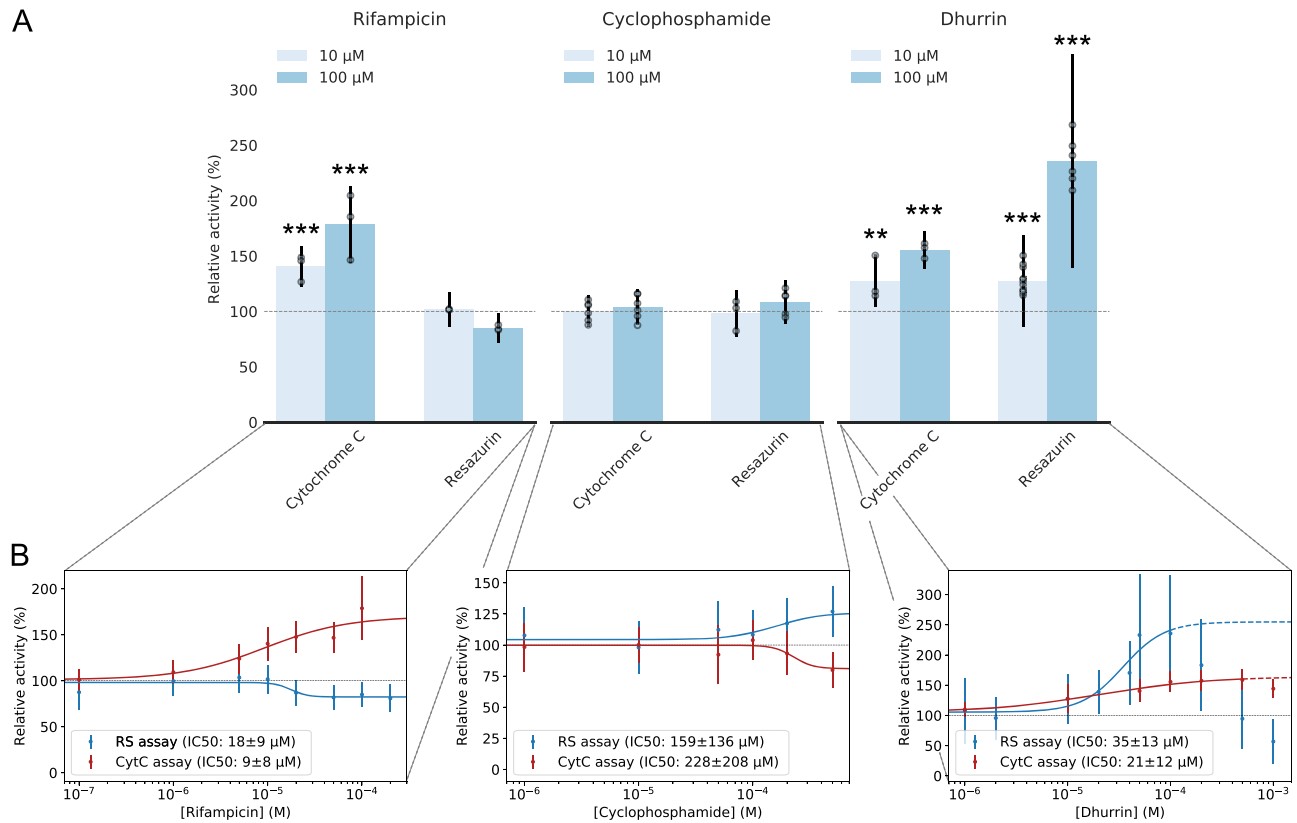

**Fig. 3 Small-molecule ligands bias specificity of plant POR (SbPOR2b) to reduce diverse electron acceptors. A** Effects of small-molecule ligands on SbPOR2b activity in proteoliposomes using Cyt*c* and RS as electron acceptors. **B** Dose-response curves of rifampicin, cyclophosphamide and dhurrin in the Cyt*c* and RS assays, respectively. Rifampicin acts as an agonist towards Cyt*c* enhancing its reduction rate and inverse agonist towards RS reducing the reduction rate. Cyclophosphamide displays the reverse effect acting as an inverse agonist towards Cyt*c* reduction and agonist towards RS reduction. Dhurrin acts as an agonist towards both Cyt*c* and RS reduction at low micromolar concentrations. The fact that ligands display differential effects on SbPOR2b activity to reduce the two electron acceptors indicates biased specificity of POR. IC50 values are extracted from the Hill equation. **A**, **B** The bar plots and dose-response curves represent the mean ± SD of independent replicates normalized to controls with propagated error ($n = 3$–12; see Supplementary Fig. 9 for raw data and Supplementary Table 3 for exact value of $n$ for each experimental condition). All data are corrected for potential ligand photophysical effect (see Supplementary Fig. 7). Note, overlapping data points appear shaded. Level of significance is determined by one-way ANOVA and Tukey's HSD test correcting for multiple comparisons (*$p < 0.05$; **$p < 0.01$; ***$p < 0.005$; see Supplementary Material for details). Source data are provided as a Source Data file.

based on intrinsic fluorescence quenching and dose-response experiments on POR2b from the crop plant *Sorghum bicolor* (*Sb*POR2b) in proteoliposomes (Fig. 3A, B, Supplementary Fig. 6 and Supplementary Table 3). Consistent with docking studies showing all three ligands binding on POR, Rifampicin increased *Sb*POR2b capacity to reduce Cyt*c* while marginally reducing the capacity in the RS assay (179 ± 35% and 85 ± 13% of control at 100 μM, respectively) with IC50 values of 9 ± 8 μM and 18 ± 9 μM, respectively (see Fig. 3B and Supplementary Fig. 9 discussing an observed lag phase). Cyclophosphamide appeared to cause the opposite effect resulting in an activity decrease in Cyt*c* reduction and increase in RS reduction (80 ± 14% and 127 ± 20% of control at 500 μM, respectively) with IC50 values of 228 ± 208 μM and 159 ± 136 μM, respectively (Fig. 3B). Note the large standard deviations due to low affinities. Michaelis–Menten kinetics of *Sb*POR in the presence of the three ligands show ligand to primarily affect $V_{max}$ supporting biased ligand binding is not competitive (see Supplementary Fig. 11 and Supplementary Methods). Careful inspection of the rifampicin and cyclophosphamide IC50 curves is reminiscent of biased agonism (Fig. 3B). Rifampicin appears to operate as an inverse agonist inhibiting RS reduction and agonist increasing Cyt*c* reduction. Cyclophosphamide, on the other hand, displays the opposite behavior and operates as an agonist towards RS reduction and inverse agonist towards Cyt*c*

reduction. Interestingly, human and plant isoforms respond differently to the same ligand, in line with differential response of GPCRs isoforms by the same ligand[33,34]. The fact that the specificity of both human and plant POR isoforms can be biased by the tested ligands, albeit to a different extent probably due to a low sequence identity of 38%, indicate that biased agonism may not be an exclusive property of receptor-mediated signaling[15] but also a method to regulate the function of metabolic cascades across kingdoms.

The natural product dhurrin caused increased *Sb*POR2b activity towards both Cyt*c* and RS (156 ± 17% and 236 ± 96% of control at 100 μM, respectively) with IC50 values of 21 ± 12 μM and 35 ± 13 μM, respectively (see Supplementary Table 3 for detailed data and replicates). The observed effect is inverted at concentrations above 100–500 μM dependent on the assay, indicating a negative feedback loop type of mechanism down-regulating dhurrin production in plants at high dhurrin concentrations (see Supplementary Methods and Supplementary Fig. 7 for control on dye photophysics). This may originate from lower binding affinity of dhurrin to alternative binding sites (see Supplementary Fig. 3 and Supplementary Table 2). The potential feedback loop may by supported by the fact that dhurrin accumulates at high (~100 mM) concentrations in sorghum[35] thus making activation of a feedback loop at these concentration

levels a likely event. Deciphering whether dhurrin is a generic regulator biasing POR-mediated CYP metabolism across all PORs is an exciting possibility that requires additional experimental verification. Ligands also have an effect on POR reconstituted in detergent micelles (Supplementary Fig. 10 and Supplementary Table 5), albeit the amplitude and precise effects are different as compared to liposomes. Similar finetuning of the effects of ligands by membrane and protein microenvironment was found for hPOR in liposomes and microsomes (see also Supplementary Fig. 8 for data on hPOR in microsomes). This is not surprising as earlier studies on POR[31,36] and additional membrane spanning or membrane associated proteins[37–39] have shown both protein dynamics and function to be altered in detergents and to be dependent on membrane properties[40–44]. POR protein and membrane microenvironment may thus finetune the effect of ligands and should be taken into consideration when analyzing POR dynamics and function. The fact that each ligand introduces diverse effects on POR capacity to reduce electron acceptors indicates that POR operates as a central metabolic hub integrating multiple layers of regulatory inputs (ligand conditions, subcellular localization, membrane environment and mutations) to tune the transferred electrons to CYPs and other redox partners, consequently controlling metabolic cascades. This opens up the possibility to use POR as a target for regulating natural product biosynthesis in plants, basic metabolism in humans, and optimize synthetic biology approaches for production of bioactive metabolites[1,11].

**Direct observation of POR conformational sampling and its remodeling by ligands**. Biased specificity is well established for receptors, and documented to operate via biased conformational sampling[15–17]. POR is a highly dynamic protein oscillating between compact and extended conformations to execute electron transfer to CYPs. This has been verified by ensemble techniques including electron paramagnetic resonance (EPR)[10], nuclear magnetic resonance (NMR)[6,9], small-angle X-ray scattering (SAXS)[6,10,45], small-angle neutron scattering (SANS)[5], fluorescence[30], and stopped-flow ultraviolet-visible (UV–VIS) spectroscopy[46], providing insights into POR conformational sampling recently confirmed by smFRET burst analysis[18,47]. Mutations known to control POR specificity and to cause metabolic disorders are often found in the hinge region of POR that controls conformational dynamics[4,12,13]. We therefore hypothesized that POR biased specificity might originate from biased conformational sampling.

We used Total Internal Reflection Fluorescence (TIRF) microscopy[48–50] to record smFRET traces and directly observe conformational sampling of SbPOR2b and its remodeling by ligands. Data were recorded using the Alternating-Laser Excitation (ALEX) methodology[51,52] that we and others have been using extensively[47,48,53]. POR was site-specifically labeled with Cy3 and Cy5 fluorophores using a minimal cysteine full-length SbPOR2b variant with two solvent accessible cysteines (N181C/C536S/A552C) that we have recently used for smFRET without impairing activity[18]. The dual-labeled SbPOR2b was reconstituted in nanodiscs, which maintain the native structure and minimize non-specific interactions with the microscope surface[18,31] (Fig. 4A and Supplementary Fig. 12A for raw images). By monitoring FRET of hundreds of single POR enzymes in parallel, we were able to quantify the conformational sampling of POR and its remodeling by the ligands (Fig. 4A). Rapid and agnostic annotation and classification of the single molecule FRET data was carried out using DeepFRET[54], our recently published methodology based on machine learning (see Supplementary Methods for details on data fitting, occupancy extraction and FRET to distance

calibration). A wide range of conformations with average FRET distances varying from ~40 to ~90 Å were observed (see Supplementary Fig. 12B for representative traces and Supplementary Fig. 13 for calibration using ALEX). Inspection of individual traces revealed relatively stable fluorescence and only rare transitions between FRET states (Fig. 4A and Supplementary Fig. 12B). This is expected as POR dynamics related to function takes place at the low millisecond time scale[6,31,46]. Imaging with a temporal resolution of 200 ms thus results in FRET states representing the equilibrium between one or more protein conformations. Indeed, decreasing the temporal resolution from 200 ms to 1 s still results in multiple distinct FRET states, however with a slightly higher fraction of traces showing dynamic transitions (from 4–8% to 10–14%; see Supplementary Fig. 13) due to longer observation times. Increasing temporal resolution to timescales faster than 200 ms was not possible without compromising signal-to-noise. Fluorescence cross correlation studies yield Pearson coefficients centered around zero for all conditions, indicating that transitions between conformational states are masked due to dynamics faster than the temporal resolution (200 ms), in agreement with our simulations and similar readouts for GPCRs[16] (Supplementary Fig. 13). Thus, the observation of multiple discrete FRET states, as well as a low fraction of transitions between them, originates from long-lived protein states as previously observed[39,55–57].

The FRET distribution of the native enzyme ($n = 243$) as well as all data combined ($n = 418$) were best fit with a mixture of five gaussians implying at least five underlying FRET states (see BIC analysis in Supplementary Methods and Supplementary Fig. 13) in agreement with earlier studies[18]. A five-state model was used to quantify the abundance of FRET states (Fig. 4B, C). In the native form, the inter-dye distances of the five FRET states were 76 Å, 64 Å, 57 Å, 48 Å and 44 Å (see Supplementary Fig. 13 and Supplementary Methods for FRET to distance calibration). We modeled the structure based on human POR in a compact conformation (PDB 3QE2[13]), rat POR in an intermediate conformation (PDB 3ES9[7]) and a human-yeast chimera in a fully extended conformation (PDB 3FJO[25]) as no crystal structure exists of SbPOR2b (see Fig. 4D and Supplementary Methods). The expected inter-dye distances calculated from dye-linker Monte Carlo simulations[58] on the three homology models were 31 Å, 59 Å and 97 Å, respectively, in agreement with our earlier burst analysis studies[18] (see Supplementary Methods), further supporting each FRET state reflects an equilibrium between multiple conformations. We call these equilibrium states S1–S5, respectively. The occupancies of the five states were 17%, 25%, 16%, 27% and 14%, respectively, implying that states S2 and S4 are the most dominant.

To evaluate the effect of ligands on POR conformational sampling, we measured smFRET on POR exposed to ligand concentrations above IC50s. Ligand binding did not significantly affect the center of the FRET states (gaussian means), but rather their occupancies, indicating a changed equilibrium between long-lived protein equilibrium states (Fig. 4B, C). Rifampicin at 100 μM shifted the equilibrium towards S1 at the expense of S4 and S5. S1 practically doubles its occupancy from 17% to 33%, while S4 decreases from 27% to 19%, but also slightly shifts its center indicating S5 slightly decreases from 14% to 11%. Cyclophosphamide was tested at 200 μM due to higher IC50 values (see Fig. 3) and was found to shift the equilibrium towards S3 (22% from 16%) and S4 (34% from 27%) at the expense of S5 and S1 (4% from 14% and 15% from 17%, respectively). Dhurrin was tested at 1 mM that appear to result in opposite responses for resazurin and Cytc reduction and was found to induce a small shift in the equilibrium between S3 and S4 (S3 increases from 16% to 22% while S4 decreases from 27% to 19%). Rifampicin

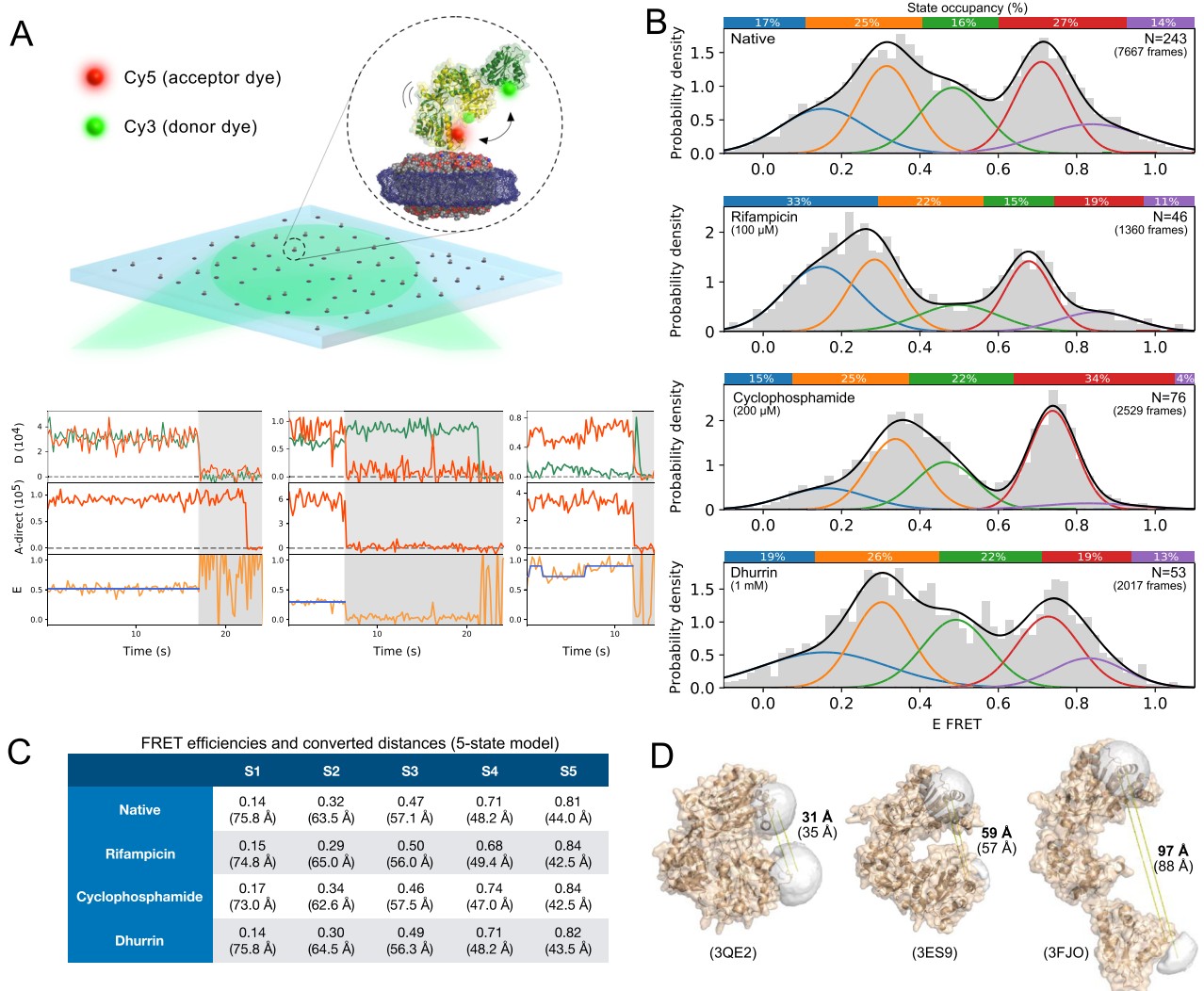

**Fig. 4 Direct observation of POR biased conformational sampling by small-molecule ligands using smFRET. A** Illustration of smFRET assay using TIRF microscopy. Top; SbPOR2b is site-specifically labeled with Cy3/Cy5 fluorophores, reconstituted in lipid nanodiscs and tethered on a passivated microscope surface. Bottom; representative smFRET traces displaying FRET states and dynamic transitions between them (see Supplementary Fig. 12 for more examples). Top row: Donor (green) and acceptor (red) intensities over time (s). Middle row: acceptor only intensity (red), bottom row: $E_{FRET}$ values (orange) calculated with calibration factors, and idealized FRET value determined from HMM fitting (blue). **B** Distribution of FRET efficiencies in the absence and presence of ligands. Distributions are optimally fit with 5 states for all conditions as determined from BIC (see Supplementary Methods and Supplementary Fig. 13) with average distances ranging from ~40 to ~80 Å. Rifampicin, cyclophosphamide and dhurrin alter the occupancies of each of the five FRET states indicating biased conformational sampling. Colored bars on top of histograms represent occupancies of each state. N denotes of the number of single molecules at each experimental condition. **C** FRET efficiencies and converted inter-dye distances obtained from five-state gaussian mixture models. Each FRET state may reflect an equilibrium between multiple conformations. **D** Homology modeling of SbPOR2b from crystal structures of POR isoforms in a compact, intermediate conformation and a human-yeast chimera in a fully extended conformation (PDBs: 3QE2, 3ES9 and 3FJO respectively) with Monte Carlo simulated inter-dye distances (bold) and Cα-Cα distances (brackets). Source data are provided as a Source Data file.

and cyclophosphamide seem to have opposed effects on conformational sampling. Rifampicin shifts the equilibrium towards extended states whereas cyclophosphamide shifts the equilibrium towards intermediate and compact states (albeit not fully compact S5). Interestingly, rifampicin and cyclophosphamide also have opposing effects on *Sb*POR2b specificity when monitored in vitro. Rifampicin operates as an agonist on Cyt*c* reduction and inverse agonist on RS reduction, while the effect of cyclophosphamide is reversed. These data thus support a correlation between conformational sampling and substrate specificity. Ligand binding on POR appears to bias conformational sampling stabilizing certain equilibrium states at the expense of others, consequently promoting the inhibition or activation of a subset of electron acceptors.

**POR ligands bias steroid hormone metabolism in human cells and microsomes**. We tested the efficiency of the three ligands to elicit a physiological response on steroidogenic CYP activities in cells (Fig. 5A, see Supplementary Methods, Supplementary Fig. 14 and Supplementary Table 4 for all data and number of biological replicates). The 3 tested ligands are not known to interact directly with CYP17, CYP19 or CYP21, albeit rifampicin and cyclophosphamide are known to interact with liver CYPs like CYP2B6[27] and CYP3A4[28]. Using a human adrenocortical cell line (NCI-H295R) we tested the effect of ligands on CYP17A1 and CYP21A2[59,60] activities at 10 and 100 μM concentrations (Fig. 5A). Using radiolabeled substrates, we were able to quantify the steroid hormone production of CYP17A1 and CYP21A2 and its remodeling by POR ligands (see Supplementary Methods for

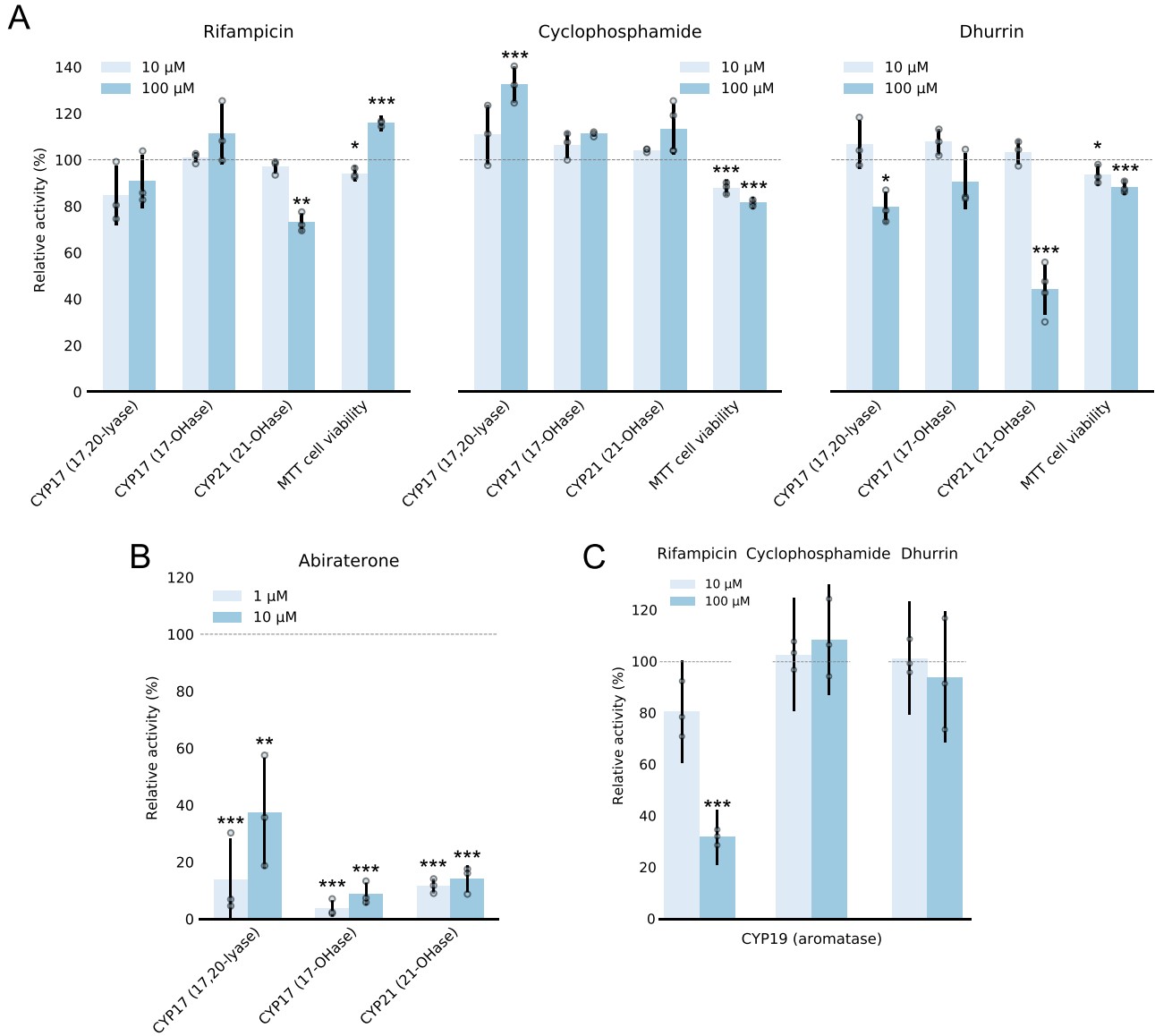

**Fig. 5 Biased metabolism: small-molecule ligands bias steroidogenic CYP-activities in human cells and microsomes. A** A human adrenocortical cell line (NCI-H295R) was used to assess the effect of small-molecule ligands on steroidogenic CYP17A1 and CYP21A2 hydroxylase activity, and CYP17A1 lyase activity, using radiolabeled substrates (see Supplementary Methods and Supplementary Table 4). Cell viability was assessed based on MTT reduction. Rifampicin shows a small inhibiting effect towards CYP21A2. The cells display increased MTT reduction indicating increased reductase activity. No significant effects on CYP17A1 activities are observed. Cyclophosphamide causes a significance increase in CYP17A1 lyase and less significant towards 17-OHase and 21A2 activities, while MTT reduction decreases slightly. Dhurrin causes inhibition of both CYP17A1 and CYP21A2 activities. **B** Abiraterone was used as a control inhibitor of CYP17A1 and CYP21A2 in H295R cells. **C** The effect of ligands on CYP19A1 activity was assessed on microsomes from a human choriocarcinoma cell line (JEG3). Rifampicin shows a concentration dependent inhibitory effect on CYP19A1 activity ($32 \pm 11\%$ of control). Cyclophosphamide and dhurrin display no significant effect. **A**, **C** Error bars represent mean ± SD of 3–4 biological replicates normalized to DMSO controls with propagated error. See Supplementary Fig. 14 for raw data and Supplementary Table 4 for exact value of n for each experimental condition. Note, overlapping data points appear shaded. Level of significance is determined by one-way ANOVA and Tukey's HSD test correcting for multiple comparisons ($*p < 0.05$; $**p < 0.01$; $***p < 0.005$; see Supplementary Methods for details). Source data are provided as a Source Data file.

details). Control experiments with the known CYP17A1 inhibitor abiraterone at saturating concentration of 10 μM[61] show specific inhibition of the two CYPs (Fig. 5B and Supplementary Fig. 14). Rifampicin caused no significant effect on CYP17A1 activity, but reduced CYP21A2 activity at 100 μM ($73 \pm 4\%$ of control; see Fig. 5A and Supplementary Fig. 14). The cells capacity to reduce MTT increased significantly ($116 \pm 3\%$ of control at 100 μM). Because MTT assay is not reliant on CYPs function, these data further support that POR docking can facilitate biased CYP

activation in cellular environment. The fact that CYP21A2 activity was reduced indicates that the observed increase in MTT reduction is not attributed to increased overall reductase expression[22], but rather remodeling of activity. Cyclophosphamide significantly enhanced CYP17A1 lyase activity and had less pronounced effect on hydroxylase activity at 100 μM ($132 \pm 8\%$ and $111 \pm 1\%$ of control, respectively), while having a small yet statistically insignificant effect on CYP21A2 activity ($113 \pm 11\%$ of control) based on t test analysis (See Supplementary

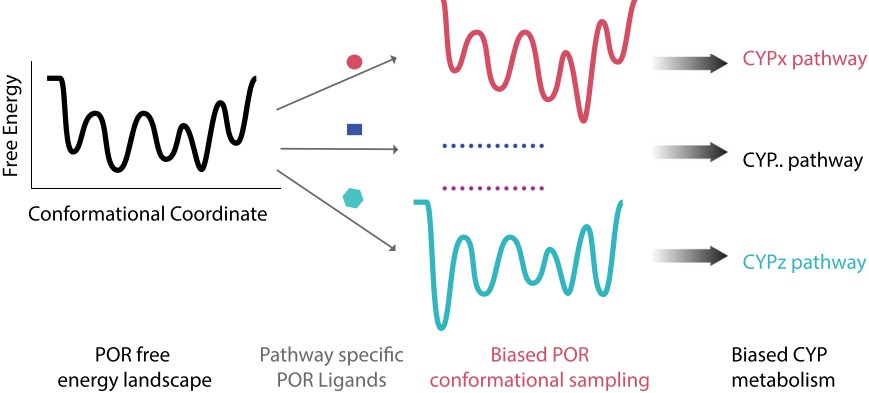

**Fig. 6 Cartoon representation of the concept of biased metabolism a mechanism akin to biased signaling of GPCRs but for metabolic hubs.** Ligand binding on POR appears to remodel its energy landscape, alter its conformational sampling consequently biasing downstream CYP activation by inhibiting the activation of a subset of CYPs and/or enhancing the activation of others. Targeting POR may act as a hitherto unknown paradigm for metabolic control in human and plants. Complete understanding of biased metabolism may offer the in silico design of pathways specific pharmaceutics or personalized food suppressing undesired, disease related, metabolic pathways.

Methods). Dhurrin caused a decrease in CYP21A2 hydroxylase and CYP17A1 lyase activities ($44 \pm 11\%$ and $80 \pm 7\%$ of control at $100\,\mu M$, respectively). Interestingly, CYP17A1 hydroxylase activity was not significantly affected by dhurrin ($91 \pm 12\%$ of control). Specific inhibition of CYP17A1 lyase activity and not hydroxylase activity, that is relevant in the treatment of prostate cancer and polycystic ovary syndrome[59], may thus be achieved by targeting POR as an alternative to targeting CYP17A1 directly. One may argue that the tested ligands may dock on additional proteins, CYPs or receptors, and induce convoluted physiological responses additional to what is reported here[22,23]. Our combined docking simulations, functional data, and smFRET structural data, illustrate that the ligands also dock on POR and affect its specificity towards reducing diverse electron acceptors.

To test the effect of ligands on CYP19A1 aromatase activity, we used microsomes extracted from a human choriocarcinoma cell line (JEG3)[60]. Radiolabeled substrate was used to quantify CYP19A1 activity and its remodeling by POR ligands (see Supplementary Methods for details). Cyclophosphamide and dhurrin did not show any significant effects at 10 nor $100\,\mu M$, while rifampicin significantly reduces CYP19A1 activity at $100\,\mu M$ ($32 \pm 11\%$ of control) (Fig. 5C and Supplementary Table 4). The microsome results further confirm that the observed effects are caused by biased activities and not via altered protein expression levels. The fact that the small-molecule ligands affect activities of steroidogenic CYPs in cells and microsomes is a key finding confirming the biological relevance of POR controlling metabolic cascades. The data confirm our MD simulations, in vitro assays and structural dynamics studies by smFRET, and support that biased conformational sampling of POR induced by ligands results in altered specificity towards CYPs. Ligand binding on POR thus appears to inhibit the activation of a subset of CYPs and/or enhance activation of others. We assign the term biased metabolism to this phenomenon since the mechanism is akin to biased signaling of receptors. We propose that biased metabolism represents an extra layer of regulatory control guiding metabolic pathways in complex cellular environments.

## Discussion

Protein conformational sampling, the dynamic exploration of conformational space, governs all major aspects of protein behavior from folding to function. Protein conformational states are often found to elicit distinct functional outcomes[55,62,63].

GPCRs are the prime example of this phenomenon, acting as key signaling hubs with several conformational states linked to distinct downstream cellular processes[15–17]. The combined studies presented here substantiates earlier studies on POR[18] and point to POR as being at the center of key metabolic hubs regulating the activation of CYPs and therefore metabolic pathways[1,3,4]. While our data do not distinguish between the swinging and rotating motion models of POR[8] they provide a correlation between the existence of POR equilibrium conformational states with distinct phenotypic metabolic outcomes.

Key advancement in our understanding of protein-ligand interactions has opened the possibility for the development of drugs that act on the same protein and selectively stabilize protein states, consequently controlling different cellular outcomes, a phenomenon described as biased agonism[15]. Biased agonism is well studied, brought into practice and explicitly exploited to underpin the function of signaling hubs like GPCRs. In our current study the combined data on the three chosen small molecules tested as ligands for POR serve as proof of concept of the mechanism of biased metabolism, a mechanism similar to biased agonism of GPCRs, but for metabolic hubs like POR. While the working concentrations ($10–100\,\mu M$) are rather high, they support the electron transfer of POR to respond to different ligands in a pluripotent way; ligand binding on POR redistributes the conformational equilibrium, consequently altering interaction with CYPs and downstream metabolism (see Fig. 6). Future studies may fully confirm each equilibrium conformational state to be directly linked to distinct downstream metabolic outcome. Biased metabolism would operate in parallel with existing and well-studied regulatory cues controlling CYP-mediated metabolism and it appears to be finetuned by variations in protein and membrane microenvironment. Interestingly, even small variations in the conformational sampling equilibrium induced by ligand binding suffices for large variations in metabolic outcomes. Thus, minute variations on ligand protein interactions, originating either by structurally diverse ligands or by different POR isoforms, may be manifested as varied metabolic responses.

Although these ligands may also interact with additional proteins[22,23] our combined in vitro, binding and functional assays, single molecule and in vivo data clearly support that their binding on POR can facilitate biased metabolism. They also highlight the significance of testing small-molecule drugs and metabolites for binding on POR during early drug discovery and high-throughput CYP screening assays[2]. We propose that biased metabolism represents an extra, hitherto unrecognized, layer of

regulation capable of controlling metabolic pathways in complex cellular environments. Targeting POR may serve as a way of controlling POR-CYP interactions and regulate CYP-mediated metabolic pathways. This is further compounded by the direct interaction of the tested ligands with amino acids where mutation is associated with POR deficiency and specifically disorder of sexual development and production of sex steroids. Our findings thus pave the way for the in silico design of pathway-specific biased ligands.

The fact that both plant and human POR isoforms, with a sequence identity of 38%, present responses to ligand binding may indicate the presence of evolutionary conserved hotspots serving to tune the specificity of POR. Indeed, sequence alignment between human, plant and rat POR reveals structurally conserved amino acids, some of which are shown by our MD simulation to interact strongly with the tested ligands (see Supplementary Figs. 2 and 5), supporting the existence of ligand binding hotspots. Additional, across kingdoms, bioinformatics analyses are required to investigate whether hotspots for ligand binding pertain across the entire spectrum of POR isoforms. Interestingly, the plant defense compound dhurrin binds on both human and plant POR and affects its dynamics, function and metabolic response in cells. Since dhurrin is a natural compound produced by a POR-CYP metabolon, these findings open up the exciting possibility of controlling biosynthetic metabolism via feedback loop mechanisms in the production of high-value natural products. Also, our results provide a mechanistic clue explaining why many substances in food, beverages and dietary supplements may affect basic metabolism and induce food-drug interactions[64,65] as dhurrin is ingested as part of foods[66,67]. It is therefore worth investigating whether additional plant or human metabolic intermediates, only present at specific developmental stages or in specific tissues, may bind to POR and bias POR-mediated metabolism.

The combined insight here may pave the way for the design of personalized, plant-based food targeting POR to alleviate metabolic disorder or dietary supplements composed of natural products to suppress specific undesired metabolic pathways associated with disease. Harnessing the structural basis of conformational sampling in biased metabolism may also offer the design of metabolic pathway-specific ligands that antagonize detrimental metabolic pathways while stimulating beneficial downstream processes, with the possibility to control basic metabolism and alleviate metabolic disorder. This could have direct implications in biomedicine for enhancing therapeutically relevant metabolic pathways. Quantitative, single molecule structural and functional studies will be crucial in this endeavor of deciphering and controlling POR-mediated metabolism via biased ligands.

## Methods

**Chemicals and materials.** All chemicals were of analytical grade and purchased from Sigma-Aldrich (Merck) unless otherwise stated. Phospholipids, 2-dilauroyl-sn-glycero-3-phosphocholine (DLPC) and 1,2-dilauroyl-sn-glycero-3-phosphoglycerol (DLPG), and 1,2-dipalmitoyl-sn-glycero-3-phosphoethanolamine-N-(cap biotinyl) (Biotinyl Cap PE) were purchased from Avanti Polar Lipids. Dhurrin was chemically synthesized in-house[1,11]. Bio-Beads SM-2 were purchased from Bio-Rad. 2'5'-ADP Sepharose size exclusion chromatography (SEC) column Superdex 200 HR 10/30 for high-resolution preparative separation were from GE Healthcare Life Sciences. Bottomless 6 channel (30 μL) sticky slides were purchased from ibidi GmbH. PLL(20 kDa) grafted with PEG(2 kDa) (PLL-PEG) and PLL(20 kDa) grafted with PEG-Biotin (3.4 kDa) (PLL-PEG-biotin) was purchased from SuSoS AG. NeutrAvidin protein was purchased from Thermo Fisher Scientific. Expression vectors were purchased from GenScript Biotech.

Radiolabeled substrates [3H]-pregnenolone, [3H]-androstenedione, [3H]-progesterone, [14C]-progesterone were obtained from PerkinElmer and American Radiolabeled Chemicals Inc. Silica gel-coated aluminum backed TLC plates were purchased from Macherey-Nagel. The tritium screens used for the autoradiography were purchased from Fujifilm. Trilostane was extracted in absolute ethanol (EtOH) from tablets commercially available as Modrenal® (Bioenvision, NY, USA). CYP17A1 was obtained from Cypex Limited. Human adrenal carcinoma cell line (NCI-H295R) and human placental JEG3 cell line was purchased from American Type Culture Collection (ATCC: CRL-2128 and HTB-36TM, respectively).

**Selection of chemical compounds for initial studies.** We used the Chemical Entities of Biological Interest ChEBI (v. 194) and downloaded the annotated subset of chemicals on 49.794 compounds as well as the approved drug subset on 2.355 compounds from Drugbank (v. 15.2). Only compounds containing C,H,N,O,P,S,F, Cl,Br,I were included. After Lipinski Rule-of-Five descriptors were calculated and compounds with a molecular weight above 1.000 removed, the datasets contained 37.708 and 2.035 compounds, respectively. The two dataset were combined and a Principal Component (PC) analysis performed based on the four Lipinski Rule-of-Five descriptors, which were normalized prior to the PC analysis (see Supplementary Fig. 1). Preparation of the datasets, calculation of the Lipinski Rule-of-Five descriptors and the PC analysis were all performed via a KNIME workflow. The 8 selected compounds were selected based on the PC analysis as structurally diverse and based on experimental and/or clinical data that could be assimilated with aberrant CYP function. Dhurrin was chosen as a representative of the family of bioactive plant natural products termed cyanogenic glycosides that are present in multiple components of the daily diet of humans. More than 3000 plant species have been shown to contain Cyanogenic glycosides. The presence of cyanogenic glycosides are five-times as common in domesticated crop species compared with plants found in natural ecosystems[66] and as such it is important to include a representative of the class from an evolutionary perspective.

**Protein expression and purification.** Full-length human wild-type POR (NCBI reference sequence: NP_000932.3 [https://www.ncbi.nlm.nih.gov/protein/NP_000932.3/]) subcloned into pET22b vectors (GenScript Biotech) was expressed in *E. coli* BL21(DE3)[3]. The cDNAs for POR in pET22b vector were transformed into the Escherichia coli BL21(DE3), single colonies were selected for growth on ampicillin and grown in terrific broth (pH 7.4) supplemented with 40 mM FeCl3, 4 mM ZnCl2, 2 mM CoCl2, 2 mM Na2MoO4, 2 mM CaCl2, 2 mM CuCl2, 2 mM H3BO3, 0.5 mg/ml riboflavin, 100 μg/ml carbenicillin at 37 °C to an optical density (OD) 600 nm of 0.6 and temperature was reduced to 25 °C for 16 h. The bacterial cells were collected by centrifugation, washed with PBS and suspended in 100 mM Tris–acetate (pH 7.6), 0.5 M sucrose, and 1 mM EDTA and treated with lysozyme (0.5 mg/ml) and EDTA (0.1 mM [pH 8.0]) at 4 °C for 1 h with slow stirring to generate spheroplasts. The spheroplasts were pelleted by centrifugation at 5000 × g for 15 min; and suspended in 100 mM potassium phosphate (pH 7.6), 6 mM MgOAc, 0.1 mM DTT, 20% (v/v) glycerol, 0.2 mM PMSF, and 0.1 mM DNase I; and disrupted by sonication. A clear lysate devoid of cellular debris was obtained by centrifugation at 12,000 × g for 10 min, and then the membranes were collected by centrifugation at 100,000 × g for 60 min at 4 °C. Membranes were suspended in 50 mM Potassium phosphate buffer (pH 7.8) and 20% (v/v) glycerol and stored at −70 °C. Protein concentration was measured by RC-DC protein assay (Protein Assay Dye Reagent, Bio-Rad, Hercules, CA). The bacterial membranes were used directly for protein reconstitution in liposomes without further purification. We note, that *E. coli* does not express any membrane-bound reductases nor CYPs which would infer with our POR activity assays.

Full-length *Sorghum bicolor* POR2b (*Sb*POR2b; NCBI reference sequence: XP_002444097.1 [https://www.ncbi.nlm.nih.gov/protein/XP_002444097.1]), subcloned into pET52b vectors (GenScript Biotech) was expressed in *E. coli* NiCo21 (DE3) cells (New England Biolabs). A 100 mL starter culture of terrific broth (TB) supplemented with 50 μg/ml ampicillin was grown overnight at 37 °C, 220 RPM. The starter culture was diluted into 1 L of TB supplemented with ampicillin in a wide bottom flask and incubated at 37 °C. Expression was induced at $OD_{600nm}$ = 0.6 by addition of IPTG to a final concentration of 1 mM. (-)-riboflavin at a final concentration of 1 μg/ml was also supplemented. Cells were harvested after expression for 6 h at 25 °C. *Sb*POR2b mutant N181C/C536S/A552C in pET52b was expressed in *E. coli* HI-Control™ BL21DE3 strain (Lucigen) in 2*400 mL TB cultures in wide bottom flasks. 1 μg/mL FMN and 1 μg/mL FAD were supplemented at the beginning of expression. Cells were harvested after 18 h expression at 20 °C by centrifugation (5000 × g, 15 min). The cell pellet was re-suspended in buffer (50 mM Tris-HCl pH 7.5, 100 mM NaCl) with 1 tablet cOmplete™ protease inhibitor cocktail (Roche) per 200 mL buffer. Cells were lysed using a cell disrupter (Constant Systems Ltd) using a process pressure of 31 kPSI. Cell debris was sedimented by centrifugation (15,000 × g, 20 min) and membranes were pelleted from the supernatant by subsequent ultracentrifugation (200,000 × g, 1 h). The membrane pellet was homogenized in buffer (50 mM Tris-HCl pH 7.5, 100 mM NaCl, 50 mM cholate) in a Potter-Elvehjem homogenizer. 100 μM FMN and FAD were also added to the solution to ensure excess cofactors. The enzymes were purified from the membrane solution using 2′5′-ADP Sepharose affinity chromatography and anion exchange chromatography according to protocols adapted from previous publications[18,31]. In brief, the membrane solution was applied to a 2'5'-ADP sepharose column equilibrated with 50 mM Tris-HCl pH 7.5, 100 mM NaCl, 20 mM cholate. After washing, bound POR was eluted with 5 mM NADP+ in buffer. Fractions containing POR were subsequently pooled and

applied to a Q-sepharose column. POR was eluted by increasing the salt concentration to 400 mM.

**Protein reconstitution in liposomes**. Liposomes were prepared using a DLPC/DLPG lipid mixture (3:1 ratio) dissolved in DMSO and placed in vacuum for ~4 h to obtain dry lipid films. While kept on ice, lipid films were rehydrated with either purified protein or bacterial membrane extract in a solution containing 50 mM cholate to obtain final lipid:protein ratios of ~200 and protein concentrations of 2–20 μM. After 1 h of incubation on a shaking platform at 5 °C, biobeads (Bio-Beads SM-2) were added to the mixture to extract detergent molecules. Upon additional 2 h of incubation, samples were centrifuged briefly to remove biobeads followed by centrifugation at $4000 \times g$ and 5 °C for 10 min. Supernatant was collected and transferred to eppendorf tubes. All samples were flash frozen and stored at −80 °C until further use.

**Intrinsic fluorescence quenching assay**. The interactions between human and plant POR and the three ligands used here (dhurrin, cyclophosphamide and rifampicin) were determined using intrinsic fluorescence quenching of aromatic residues. Plant SbPOR2b in detergent (4.6 μM) was prepared in 50 mM Tris-HCl (pH 7.5) containing 20 mM cholate and 100 mM NaCl, while hPOR in microsomes was prepared in PBS buffer (pH 7.2) containing 10% glycerol. The fluorescence quenching was measured on a fluorescence spectrometer (SpectraMax M2e, Molecular Devices, CA USA) at room temperature. The samples were excited at 295 nm and emission was detected between 300 and 500 nm. Tryptophan fluorescence spectra were collected before and after addition of 100 μM of each of the ligands. All data are corrected for negligible ligand fluorescence changes under the same experimental conditions. Note that intrinsic fluorescence quenching provides limited information about which tryptophan or even tyrosine is quenched, and thus whether ligands dock on site Ia or/and site Ib. The observed significant quenching fully supports that all three ligands dock on POR.

**In vitro POR activity assays**. The activity of full-length POR reconstituted in either detergent micelles (20 mM cholate) or liposomes (DLPC/DLPG, 3:1 ratio) in 50 mM Tris-HCl buffer (pH 7.5) containing 100 mM NaCl was assessed spectrophotometrically at 10–100 nM concentrations as described elsewhere[29–32]. Using Cyt$c$, MTT or RS as electron acceptors the change in absorbance (550 nm for Cyt$c$ and 610 nm for MTT) or emission (570 nm excitation, 585 nm emission for RS) was monitored as a function of time. Cyt$c$ and RS concentrations were both close to $K_M$ of the given electron acceptor, while MTT and NADPH were in excess amounts. Concentrations were 40 μM Cyt$c$, 10 μM RS, 500 μM MTT and 100 μM NADPH unless otherwise stated. POR activity was extracted from the slope of the linear region of each trace. Michaelis–Menten kinetics and IC50 curves were performed under identical conditions using relevant substrate and ligand concentrations. All measurements were repeated at least three times and subsequently normalized to DMSO control measurements. All data were background corrected and controls without POR showed negligible activity. Fluorescence intensities of resorufin in the presence of rifampicin were corrected according to a linear calibration curve (see Supplementary Fig. 7).

**hPOR dose-response activity assays in microsomes**. In a typical experiment hPOR microsomes (100–500 ng/well) extracted from bacterial membrane was allowed to react with 40 μM Cyt$c$, 10 μM RS or 500 μM MTT in 100 mM phosphate buffer (pH 7.2) in the presence of 100 μM NADPH and increasing concentration of ligands (dhurrin, cyclophosphamide and rifampicin). All data were recorded with a Spectramax M2e spectrophotometer (Molecular Devices, Sunnyvale, CA, USA) by measuring the change in absorbance over time (550 nm for Cyt$c$ and 610 nm for MTT) or emission (570 nm excitation, 585 nm emission for RS). Reactions were started by addition of NADPH at 100 μM. See Supplementary Fig. 8 for data.

**Cell lines and culture media**. Cells were cultured according to established protocols[59,60]. Human placental JEG3 cells were cultured in minimal essential medium (MEM) with Earle's salts (Thermo Fisher Scientific) supplemented with 10% fetal bovine serum, 1% L-glutamine (200 mM GIBCO), 1% penicillin (100 U/ml; GIBCO), and streptomycin (100 μg/mL; Thermo Fisher Scientific). Human adrenocortical NCI-H295R (NCI-H295R) cells were grown in DMEM/Ham's F-12 medium containing L-glutamine and 15 mM HEPES (Thermo Fisher Scientific) supplemented with 5% NU-I serum (Becton Dickinson), 0.1% insulin, transferrin, selenium (100 U/mL; Thermo Fisher Scientific), 1% penicillin (100 U/mL; Thermo Fisher Scientific), and streptomycin (100 μg/mL; GIBCO) and passage numbers during the experiments remained below 30.

**Preparation of microsomes and CYP19A1 activity assay using JEG3 cells**. Microsomes were prepared from JEG3 cells based on a protocol adapted from previous work[60]. JEG3 cells were collected near confluency and washed with cold PBS. The cell suspension was then centrifuged at $1500 \times g$ for 5 min to pellet the cells. The cell pellet was suspended in 100 mM $Na_3PO_4$ (pH 7.4) containing 150 mM KCl, and the cells were lysed by sonication. Unbroken cells and mitochondria were pelleted by centrifugation at $14,000 \times g$ for 15 min at 4 °C.

Microsomes containing endoplasmic reticulum were collected by ultracentrifugation at $100,000 \times g$ for 90 min at 4 °C and resuspended in 50 mM $K_3PO_4$ (pH 7.4) containing 20% glycerol.

CYP19A1 activity was measured by the release of tritiated water from radiolabeled substrates during aromatization[60]. First, 40 μg microsomes extracted from JEG3 cells were incubated with 50 nM [$1_β$-$^3$H(N)]-androstenedione (~20,000 cpm/reaction) in buffer (100 mM NaCl, 100 mM potassium-phosphate, pH 7.4) for 5 min at 37 °C on a shaking platform. The reaction was initiated by adding 1 mM NADPH. After 1 h of incubation at 37 °C, the reaction was stopped by adding a mixture of 5% charcoal and 0.5% dextran. Samples were vortexed for 40 s and centrifuged at $14,000 \times g$ for 5 min. Supernatant was collected and diluted in scintillation liquid (Rotiszint Universal Cocktail; Carl Roth GmbH) before counting [$^3$H]-radioactivity. All measurements were repeated at least three times and subsequently normalized to controls.

**CYP17A1 and CYP21A2 activity assay in H295R cell line**. Steroidogenic CYP17A1 and CYP21A2 activities in H295R cells were quantified based on radiolabeled substrate assays[59,60]. In brief, cells were plated in six-well plates and treated with small-molecule ligands dissolved in 0.1% DMSO and normal growth medium for 24 h, or for 4 h for rifampicin, to ensure protein expression levels are not affected[22]. After incubation, 1 μM trilostane (a specific blocker of HSD3B) was added to the medium for 90 min followed by addition of 1 μM radiolabeled substrate ([$^3$H]-17α-OH-progesterone or [$^3$H]-pregnenolone; ~50,000 cpm). After 1 h of reaction, steroids were extracted from cell supernatants and separated by thin layer chromatography (TLC) on silicagel (SIL G/UV$_{254}$) TLC plates (Macherey-Nagel, Oensingen, Switzerland). The steroids were visualized on a Fuji FLA-7000 PhosphorImager (Fujifilm, Dielsdorf, Switzerland) and quantified using Multi Gauge software (Fujifilm, Dielsdorf, Switzerland).

Steroid conversion was assessed as a percentage of incorporated radioactivity in relation to total radioactivity measured for the whole sample. The conversion of 17α-OH-progesterone to 11-deoxycorticosterone was used as a measure for CYP21A2 hydroxylase activity. The conversion of pregnenolone to 17α-OH-pregnenolone and dehydroepiandrosterone (DHEA) was used as a measure for CYP17A1 hydroxylase activity, while the specific conversion of 17α-OH-pregnenolone to DHEA was used as a measure for 17,20-lyase activity. All measurements were repeated at least three times and subsequently normalized to controls.

**MTT cell viability assay**. MTT reduction was used to evaluate cell viability and simultaneously quantify reductase expression/activity of cells upon drug incubation as described elsewhere[60]. In brief, 100 μL of cell solution containing ~$3 \times 10^4$ cells were placed in a 96-well plate at a concentration of upon 24-h incubation with drugs. MTT was added in each well to a final concentration of 0.8 mg/mL. Absorbance was measured at 610 nm on a plate reader to quantify reduction of MTT. All measurements were done in triplicates and normalized to DMSO controls.

**Homology modeling and Monte Carlo simulation of dye-dye distances**. Since no 3D structure of SbPOR2b is available, the structure was modeled using SWISS MODEL automated online server[68] based on human, yeast and rat POR isoforms[18]. The compact conformation of SbPOR2b was modeled based on human POR (PDB 3QE2[13], chain A) while the intermediate and fully extended conformations were based on rat POR (PDB 3ES9[7], chain A) and a human-yeast chimera (PDB 3FJO[25]), respectively. The isoforms share 38%, 40% and 36% sequence identity to SbPOR2b, respectively.

To convert the modeled Cα-Cα distances between residues N181 and A552 to expected dye-dye distances as measured by smFRET we used a toolkit developed by Kalinin et al.[58] based on Monte-Carlo simulations. The toolkit employs a geometric accessible volume (AV) algorithm to predict the spatial distribution of donor and acceptor dyes. Using an approximated linker length of 14.0 Å, linker width of 4.5 Å and dye radius of 3.5 Å, the simulated dye-dye distances and modeled Cα-Cα distances differ by 3.7–8.9 Å depending on protein conformation (see main Fig. 4, panel D). The 14.0 Å long linker length ensures free rotation of the dyes, which is essential for distance calibration.

**Computational docking simulations on POR structures**. The small molecules were constructed in Maestro (v. 9.8, Schrodinger 2018-3 release, Schrödinger, LLC, New York, NY, 2014) and prepared for docking in the LigPrep module[69]. Both neutral and charged forms were generated and used as input structures for the docking. The rifampicin structure was taken from a structure of rifampicin monooxygenase complexed with rifampicin (PDB 5KOX[70]). Prior to docking, rifampicin was subjected to the LigPrep module, which produced three different conformations. After a short energy minimization these structures were used as input structures for the docking.

Identification of potential binding sites was carried out using the SiteMap software (v. 2.6, Schrödinger, LLC)[24]. All ligands were docked on the human POR crystal structure (PDB code 3QE2[13], A-chain) as well as the rat POR crystal structure (PDB code 3ES9[7], A-chain) representing the compact and extended POR conformations, respectively, using Glide (v. 5.8, Schrödinger, LLC) with default

settings (van der Waals scaling factor and partial charge cut-off at 0.80 and 0.15, respectively; standard precision and flexible ligand sampling; no constraints)[71,72]. All docking results are displayed in Supplementary Table 2.

**MD simulation to evaluate the stability of top scoring docking conformations**. The stability of top scoring docking conformations was evaluated to be stable by short MD simulations (embedded in water box, 10 ns) in Desmond (v. 4.3, Schrödinger, LLC)[73]. The Desmond system builder was used to create an orthorhombic box with the protein-ligand complex embedded in an SPC water model with a buffer size of at least 10 Å from the protein to the box boundary. The final systems contained close to 72,000 atoms with ~9600 atoms from the protein, 200 atoms from cofactors and ligand, 62,300 atoms from water molecules and nearly 30 sodium ions to neutralize the systems.

Extended 1000 ns MD simulations were performed with the Desmond program using the OPLS-2005 force field. Prior to the production simulations, the systems were subjected to the Desmond standard stepwise equilibration procedure comprising a restrained energy minimization of the solute, a 12 ps simulation at 10 K on an NVT ensemble, a 12 ps simulation at 10 K on an NPT ensemble, and a 12 ps simulation at room temperature on an NPT ensemble. Finally, the systems were equilibrated without constraints for 24 ps at room temperature on an NPT ensemble. After equilibration the systems were simulated for 1000 ns at 300 K and 1000 frames were collected and analyzed. Key parameters, Cα RMSD values and protein-ligand contacts, for the MD simulations are collected in Supplementary Fig. 5.

**Protein labeling and nanodisc reconstitution for smFRET**. *Sb*POR2b mutant N181C/C536S/A552C containing two solvent accessible cysteines was labeled with Cy3 and Cy5 maleimide mono-reactive dyes (GE Healthcare) according to established protocols[18]. In brief, 1 mg *Sb*POR2b was incubated with 200 μM DTT at 4 °C for 2 h. Cy3 and Cy5 solubilized in DMSO was added to POR and incubated over night at 4 °C. A stoichiometric ratio of 1:2 Cy3/Cy5 was used for optimal labeling conditions. Control samples labeled with one type of fluorophore were also included. Samples with fluorophores were kept dark to prevent photobleaching. Free dye was separated from labeled POR by size exclusion chromatography on PD-10 desalting columns with Sephadex G-25 resin (Sigma Aldrich) equilibrated with buffer (50 mM Tris-HCl pH 7.5, 100 mM NaCl, 20 mM cholate). Labeled protein was reconstituted in lipid nanodiscs comprising membrane scaffold protein MSP1E3D1 and a mixture of DLPC:DLPG:Biotinyl Cap PE:DiO (69.8:25:4:1.2 ratio) according to established protocols[18,31]. In brief, detergent solubilized POR was mixed with nanodisc constituents, 5 mM DTT and 100 μM FAD and FMN. The mixture was incubated on ice on a shaking platform for 1 h followed by addition of biobeads and additional 4 h of incubation facilitate detergent removal. Samples were centrifuged briefly to remove biobeads followed by centrifugation at 4000 x *g* and 5 °C for 10 min. Supernatant was collected and transferred to eppendorf tubes. Purification of nanodiscs was achieved by size exclusion chromatography (SEC) (flow rate: 0.5 mL/min) on a preparative HPLC (Shimadzu) equipped with a Superdex 200 Increase 10/300 GL column (Amersham Pharmacia Biotech; diameter 10 mm; length 300 mm) using a mobile phase of 50 mM Tris-HCl (pH 7.5), 100 mM NaCl. Elution of protein, Cy3 and Cy5 was continuously monitored by absorbance at 280, 550 and 650 nm, respectively. Selected fractions of nanodiscs containing POR were collected, flash frozen and stored at −80°C until further use.

**Surface preparation and nanodisc immobilization for smFRET**. Dual-labeled POR reconstituted in lipid nanodiscs was immobilized on a PLL-PEG functionalized surface. The surfaces were prepared according to established protocols[31,48]. In brief, glass coverslips were cleaned, dried under nitrogen flow and plasma etched (Harrick Plasma Cleaner PDC-32G-2) for 5–10 min at 60 Pa. Immediately after plasma etching, flow chambers were assembled using 6-channel sticky slides (Ibidi) and each chamber was incubated with a 1:100 mixture of PLL-PEG-biotin/PLL-PEG in HEPES buffer (pH 5.6) for at least 1 h. Chambers were flushed with 1 mL buffer to remove excess PLL-PEG and afterwards incubated with 0.1 g/L NeutrAvidin for at least 10 min. The chambers were stored at 5 °C until further use. Prior to each measurement, excess NeutrAvidin was removed by flushing each chamber with ~1 mL buffer followed by incubation with lipid nanodisc for at least 5 min to ensure immobilization. Excess nanodiscs were removed by flushing with buffer.

**Acquisition of smFRET data**. All smFRET experiments were performed on a total internal reflection fluorescence (TIRF) microscope (IX83, Olympus) equipped with two EMCCD cameras (imagEM X2, Hamamatsu) and an oil immersion 100x objective (UAPON 100XOTIRF, Olympus). Cy3 and Cy5 fluorophores were excited with 532 nm and 640 nm solid state laser lines (Olympus), respectively. A quad band filter cube was used to block the lasers in the emission pathway, while a multichannel imaging system (DC2 two-channel system, Photometrics) was used to split the signal into two channels. 582/75 nm and 700/75 nm band pass ET filters were used to filter donor (Cy3) and acceptor (Cy5) emission, respectively. All experiments were performed using ALEX methodology[48,51] with 200 ms temporal resolution. All experiments were done in imaging buffer (50 mM TRIS, 600 mM

NaCl, pH 7.9) with triplet quenchers (2 mM trolox, 2 mM para-nitrobenzyl alcohol, 2 mM cyclooctatetraene) and an oxygen scavenging system (1 U/mL PCD, 2.5 mM PCA) to avoid blinking and improve photostability of the fluorescent dyes.

**Quantitative analysis of smFRET data**. Quantitative image analysis and classification of smFRET traces was carried out using DeepFRET[54], our recently published methodology based on machine learning. Time-dependent signal and background traces were extracted for each colocalized Cy3/Cy5 pair and classified by the Deep Neural Network model encompassing features like signal/background ratio, noisiness, photobleaching and stoichiometry. Extracted traces were sorted by setting a user-defined quality threshold to ensure only traces corresponding to single donor and acceptor fluorophores with anti-correlated signals were used for further analysis.

To correct FRET values for spectral crosstalk, direct excitation and detection efficiency, DeepFRET implements the method precisely outlined by Hellenkamp et al.[52] based on established methodology as first described by Lee et al.[51]. In brief, we obtained α and δ correction factors for spectral crosstalk and direct excitation by manual inspection of the traces. Afterwards, we obtained the β and γ factors from FRET versus stoichiometry 2D histograms. The corrected FRET efficiencies were converted to distances using a Cy3/Cy5 Förster radius of 56 Å.

The distributions of FRET values for each experimental condition were fitted with a mixture of gaussians using unbinned likelihood fitting. Although FRET values are not technically Gaussian distributed, it has in practice been shown to be a robust method with little discrepancy[16,47,48]. The number of underlying states (i.e., gaussians) was determined based on Bayesian Information Criterion (BIC). Since BIC is highly sensitive to the number of data points and each experimental condition had a different number of accepted FRET traces, the number of underlying states was determined from the largest data set as well as all data combined both resulting in five gaussians as the best model (Supplementary Fig. 13). The fit outputs the mean, sigma and weight of each gaussian within the gaussian mixture. Based on this, even though the distributions overlap, the actual contribution of each state to the overall distribution is accurately extracted. Hidden Markov modeling (HMM) was used to identify dynamic transitions[48].

**Simulation of FRET traces and cross-correlation analysis**. Traces were generated based on a two-state hidden markov model (HMM) with a transition probability of 0.95 between FRET states at 0.3 and 0.7, respectively. 8% gaussian noise was added to donor and acceptor intensities to simulate experimental uncertainty. To mimic various temporal resolutions, traces were binned at 1, 5, 10, 20, 50 and 100 frames per bin effectively causing between 0.95 and 95 transitions on average per bin at the various conditions. After binning, all traces were cut to a length of 200 frames with no bleaching to ensure equal amounts of data at different temporal resolutions. The Pearson cross-correlation was calculated for donor and acceptor intensities of each trace subsequent to binning.

**Statistical analysis of in vitro activity data**. The average and standard deviation (SD) of at least three independent replicates were calculated followed by normalization to the average and SD of at least three independent control measurements without ligand. The effect of each ligand was reported as a percentage of the control with error propagation. All uncertainties represent ±SD unless otherwise stated. *P* values were calculated based on *t*-tests correcting for multiple comparisons, since the effect of multiple ligands were compared to the same set of control measurements. For each set of control experiments, a one-way ANOVA test was performed to decide whether the effect of one or more ligands were significantly different from the control (confidence level: $p < 0.05$). In case of a statistically significant difference, Tukey's HSD test was used for pairwise comparison between each of the ligands and the control to determine which of the ligands had a significant effect while correcting for multiple comparisons. The level of significance as determined by Tukey's HSD test is marked by asterisk symbols on the corresponding bar charts (*$p < 0.05$; **$p < 0.01$; ***$p < 0.005$).

**Fitting of dose-response (IC50) curves**. Dose-response curves from POR activity assays were fitted using the Hill equation in order to determine the IC50 value:

$$f(x) = A + \frac{B - A}{1 + \left(\frac{IC50}{x}\right)^n}$$

Here, A represents the starting point (no ligand), B represents the end point (maximum response) and n represents the Hill coefficient. The Hill coefficient was bounded to the interval [0,5] in the presence of rifampicin and cyclophosphamide. Dhurrin IC50 values were extracted from lower concentrations only (solid lines in Fig. 3B) as the effect is inverted at concentrations above 100–500 μM dependent on the assay. This is not due to photophysical effects (Supplementary Fig. 7). The activity loss at high dhurrin concentrations may indicate the presence of a feedback loop where high dhurrin concentrations inhibit POR electron transferring activity. This is especially important as POR is crucial for dhurrin formation and indicates a self-regulatory cue inhibiting POR function when high concentrations of dhurrin are produced. While the mechanism for this is unclear, it could originate from an alternative binding site with lower affinity. Deciphering this, however, extends beyond the scope of this work and is worth further investigation in future studies.

**Reporting summary**. Further information on research design is available in the Nature Research Reporting Summary linked to this article.

## Data availability

All relevant data are displayed in the main figures or the supplementary materials. Source data are provided with this paper and deposited at UCPH ERDA server accessible via the link: https://sid.erda.dk/sharelink/EpwrxUSWOe

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

## Acknowledgements

This work was supported by the Carlsberg Foundation Distinguished Associate Professor Program (CF16-0797), the VILLUM Foundation Young Investigator Program (10099) and the VIllum foundation center of excellence BIONEC (18333) to N.S.H. and by the VILLUM Center for Plant Plasticity (VKR023054) headed by B.L.M., by a European Research Council Advanced Grant (ERC-2012-ADG_20120314), a Novo Nordisk Foundation Distinguished grant (NNF19OCOO54563), and a Novo Nordisk Foundation Interdisciplinary Synergy grant (NNF16OC0021616) to B.L.M. NSH is a member of the Integrative Structural Biology Cluster (ISBUC) at the University of Copenhagen and associate member of the Novo Nordisk Foundation Center for Protein Research, which is supported financially by the Novo Nordisk Foundation (NNF14CC0001). A.V.P. was supported by grants from the Swiss National Science Foundation (31003A-134926) and the Novartis Foundation for Medical-Biological Research (18A053). T.L. and R.D.G. were supported by a Sapere Aude Starting Grant from the Independent Research Fund Denmark (7026-00041B) and a fellowship awarded by the Novo Nordisk Foundation (NNF17OC0024886). Authors thank FIDA-Tech Aps for fruitful discussions and use of "Fida 1 platform" for preliminary ligand binding assays of intrinsic fluorescent quenching based on Flow Induced Dispersion Analysis.

## Author contributions

S.B.J. performed all in vitro activity assays, smFRET assays and data analysis with the help of J.T., P.M.L., Y.G.B., M.B.S. and M.N.R.V. S.B.J. performed homology modeling, Monte Carlo simulations and simulation of smFRET traces for cross correlation studies with the help of J.T. and M.B.S. Protein expression, purification and liposome reconstitution was performed by S.T., C.C.H., C.K., R.D.G. with help of S.B.J. and M.E.M. Labeling of *Sb*POR2b and reconstitution in nanodiscs was performed by S.B.J., C.C.H., S.T., M.E.M. and T.L. Cell assays were performed by S.P. and P.R.C. with help of S.T., M.E.M. and S.B.J. F.S.J. performed docking simulations. S.B.J. and N.S.H. wrote the manuscript with inputs from all authors. N.S.H. conceived the research initiative, had the overall strategic planning with help of B.L.M. and together with B.L.M. were responsible for the overall project supervision. All authors discussed and evaluated all data.

## Competing interests

The authors declare no competing interests.
