## [Peer Review File · Nature Communications]

REVIEWER COMMENTS

Reviewer #1 (Remarks to the Author):

The paper entitled "Across kingdom biased CYP-mediated metabolism via small-molecule ligands docking on P450 oxidoreductase" describes: i) how ligands can bind specifically to NADPH cytochrome P450 reductase (POR), ii) how this binding modifies the conformational equilibrium landscape of POR and, iii) in return, how the changes in the conformational equilibrium finally affect differentially electron transfer (ET) from POR to various acceptors. The ultimate claim of the paper is that ET from POR to cytochromes P450 (CYPs) can be biased by ligands.

The conclusions of the paper are novel and original and extremely important. The fact that CPR may be biased toward certain CYPs has already been evidenced both in reconstituted systems with variant forms of CPR and from genetic data on CPR polymorphism, leading to the hypotheses that control of specificity by POR can be triggered by either a change in the conformational equilibrium or by mutations that induce redox partner selection. The present article adds a truly additional and original mechanism for specialization: binding of molecules at the interface between the FMN and FAD domains of POR leads to changes in the conformational equilibrium only, which, in return, potentiates certain redox partners interactions while reducing others.

The discovery of this phenomenon has strong implications on drug metabolism: molecules tested here are known to be substrates or inhibitors of human CYPs related to drug metabolism. The results presented here demonstrate that these small compounds induce a modulation of CYPs activities while not being linked to CYPs binding per se, leading to the potential "reuse" of drugs already on the market for other purposes and also potentially explaining adverse drug effects that were not related to CYP direct activation or inhibition.

The claims of the paper are of strong interest to readers outside of the POR/CYP interaction fields because it demonstrates that conformational equilibrium changes within one protein can lead to specific protein-protein interactions and preferential metabolism. On a general point of view, the paper explains the molecular basis for "allosteric metabolic control" by proteins at the centre of metabolic hubs. This point of view is quite original and gives mechanistic and atomistic explanation of metabolon controls.

Several points merit attention prior publication.

Docking of POR ligands induce biased specificity towards electron acceptors.

Line 89 and further. How these ligands were chosen? In particular, the selection includes molecules that are gold standards for CYP assays. Did the authors have a preliminary screen or did they test all possible drugs? If a preselection was performed, was it done with docking or functional activity assays? Did the authors use a threshold for MW before (potential) screening of the ligands? A small comment on the library that was screened would suffice but it would be quite interesting to know the proportion of ligands that bind POR (docking) within a library of small molecules which have been described to affect CYP activities and, amongst the ones that dock, which proportion is effectively active on POR. This is of particular interest as these molecules are also substrates and/or inhibitors of CYPs themselves. Finally, the use of dhurrin as a test molecule for mammalian POR is surprising. Of course, some of the authors are specialists of the dhurrin pathway for which metabolic control has been demonstrated, but what was the rationale to test it with human POR? Would dhurrin be a general regulator of all POR?

Line 92 and further. In some cases, molecules were not tested at the second concentration (100 μ M). Why? Maybe the authors should explain this in the figure caption. Why did the authors report the SEM instead of the SD? Even if the calculation of the SD from the SEM is quite straightforward, reporting the dispersion of the data points is much more informative than the accuracy of the mean. This is especially true with cyclophosphamide (in Figure 2D) for which the SEM is greater than 1/3rd of the mean value and therefore the SD is greater than 50% of the mean value. This question holds true for the rest of the paper evidently except for Fig. S5 where SD are actually reported.

Line 98 and further. The extended conformation is from rat POR. As stated in the Supp Fig. 2, Sites IV on human POR do not completely align with Sites I-V on rat POR. Why the authors did not use an extended conformation for human POR, as they did for SbPOR? It would be more logical to analyse the

binding on the compact or extended conformations with the same sets of amino acids, even if the identity between human and rat POR is 94%. This could even help get better precision on how the binding affects or is affected by the transition from the compact to the extended form. Line 126 and further. "All compounds were tested at 100 μ M using 40 μ M Cytc and 100 μ M NADPH as substrates". In the SI, the authors stated that: "Cytc and RS concentrations were both close to K_m of the given electron acceptor, while MTT and NADPH were in excess amounts". Testing at the K_m value is not necessarily a good idea as it is difficult to sort the effect of the ligands on the saturation behaviour or the k_{cat} . A full saturation curve of Cytc activity with the four different molecules would be more informative, especially if either the K_m or the k_{cat} or both are modified. In Figure 2, in the Cytc assay: what was the concentration of CPR used in the assay (approximately?). The relative absorbance change is rather small in 100 seconds: approximately 0.001 OD in 1 min, yielding approximately 50 nM of reduced Cytc per minute for Cyclophosphamide. This tends to indicate that POR was present at very low concentration in the test, between 50 to 100 pM, if not lower. At these concentrations, POR loses FMN very fast if in the oxidized form prior to NADPH addition. The potential loss of FMN is evidently critical to accurately measure POR activity. In the referenced papers (SI, ref 24-27) the buffer conditions for POR activity tests are different (Ref 24 is using 50 mM KPO₄ buffer, Ref 25 is using 50 mM Tris and various concentrations of NaCl, Ref 26 is using 100 mM KPO₄ buffer and Ref 27 50 mM Tris). It would be more helpful to precisely describe the conditions of the tests, in particular the salt concentration, buffer used and concentration of POR.

POR ligand binding and biased specificity pertain across kingdoms.

Line 183 and further. The authors present the full curves (IC₅₀) for inhibition and activation. This data is much nicer than just the end points presented in the previous section. Why the authors did not use it with human POR. Only IC₅₀ curves can evidence mixed-type behaviours such as dhurrin with Cytc or RS.

Line 203 and further. The sentence "The observed effect is inverted at concentrations above 100-500 μ M dependent on the assay, indicating a negative feedback loop type of mechanism downregulating dhurrin production in plants at high dhurrin concentrations" is a bit misleading as the authors describe the effects on alternate electron acceptors. The data presented throughout the article tend to demonstrate that the stimulating or inhibiting effects of the molecules tested are electron acceptor dependent. Therefore, the authors cannot ascertain that the dhurrin effect observed on Cytc and RS is identical with CYPs participating in dhurrin synthesis in sorghum (although this is a very good hypothesis that could be easily tested with microsomal fractions). The sentence "Although the amplitude and effects are different in detergent compared to liposomes, it highlights that the observed effects are not induced by altered properties of the lipid bilayer" seems counterintuitive. On the contrary, the observed differences can demonstrate that part of the effects may be attributed to altered properties of the lipid bilayer. Otherwise the effects, if they have absolutely no dependency on the immediate physicochemical environment of POR, should be identical unless the authors think that POR ET properties (as well as dynamics etc.) are different in proteoliposomes vs. detergent. This point should be discussed more extensively, in particular the potential role of the microenvironment of POR (as this is also mentioned further in the paragraph).

Direct observation of POR conformational sampling and its remodeling by ligands

Line 276 and further. The authors should comment on the difference between the calculated distance using FRET in the compact form (44 Å) and the one obtained by modelling (31 Å): 13 Å seems quite a lot, especially for the compact form. Are the dyes positions fixed?

Line 291 and further. "To evaluate the effect of ligands on POR conformational sampling, we measured smFRET on POR exposed to ligand concentrations well above IC₅₀". The rifampicin concentration used (100 μ M) is effectively well above the EC₅₀. However, for cyclophosphamide, 200 μ M is not at all "well above" EC₅₀ (159 and 228 μ M). For dhurrin, it would have been very informative to add another concentration, the one where SbPOR activated RS reduction (100 μ M of dhurrin) to analyse it in view of the effect at 1 mM where SbPOR reduces RS reduction.

POR ligands bias steroid hormone metabolism in human cells and microsomes

Line 327 and further. "Rifampicin caused no significant effect on CYP17A1 activity, but reduced CYP21A2 activity at 100 μ M (73 \pm 2% of control). Considering the SD and not the SEM, the decrease of activity of CYP17A1 with rifampicin at the two concentrations and not be so different from the effect at

100 μ M rifampicin for CYP21A2. "Cyclophosphamide enhanced both CYP17A1 hydroxylase and lyase activity at 100 μ M (111 \pm 1% and 132 \pm 5% of control, respectively), while having a small yet statistically insignificant effect on CYP21A2 activity (113 \pm 5% of control)". I believe the authors wanted to write significant instead of insignificant? In any case, if the 111 \pm 1% is considered significant for the authors, 113 \pm 5% is certainly significant as well!

Overall, in this part, one control is missing: the absence of direct interactions of rifampicin, cyclophosphamide and dhurrin with CYP17A1, CYP21A2 and CYP19A1. Rifampicin and cyclophosphamide are known to interact with several drug-related CYPs. The probability that they interact with the three abovementioned CYPs is not small and should be assessed, directly on microsomal fractions, by spectral changes (Type I or Type II). For dhurrin, the control should be performed as well.

Minor points:

Main text:

Line 49: the generalization is misleading. Metabolic control does not only apply to redox protein partners and cytochromes P450, the latter only represent part of the metabolism.

Line 67: its instead of Its (after the colon).

Line 69: the reference 19 describes how mutations in the FAD domain can switch specificity of POR from NADPH to NADH and, as a consequence, induce a differential coupling with human CYPs. Other articles that also describe specificity changes due to mutations on POR should be cited to be completely exhaustive: Esteves, F. et al. The Role of the FMN-Domain of Human Cytochrome P450 Oxidoreductase in Its Promiscuous Interactions With Structurally Diverse Redox Partners. *Front Pharmacol* 11, 299 (2020). Campelo, D. et al. Probing the Role of the Hinge Segment of Cytochrome P450 Oxidoreductase in the Interaction with Cytochrome P450. *Int J Mol Sci* 19, (2018).

Line 127: comma between activity and support not necessary.

Line 241: The SAXS reference number 6 is incomplete: the reference number 9 and another reference (Huang, W.-C., Ellis, J., Moody, P. C. E., Raven, E. L. & Roberts, G. C. K. Redox-Linked Domain Movements in the Catalytic Cycle of Cytochrome P450 Reductase. *Structure* 21, 1581–1589 (2013)) should be added to be completely exhaustive.

Supplementary Information:

Line 290: 3ES9 instead of 3ES.

There is a general problem with the reference numbering in the SI. For example, reference 2 (line 37) in the SI is probably wrong; line 70 refs 24-27 possibly wrong as well (not all of them); line 81 ref. 42; line 92 ref. 42; line 101 ref. 42; line 112 ref. 42; line 133 ref. 42; line 141 ref. 47 (which does not exist in the reference section); line 148 ref. 40; line 156 and 159, ref. 48 and 49 are not present in the reference section, etc. Please recheck all references in the SI.

Reviewer #2 (Remarks to the Author):

The Authors propose a concept of biased metabolism, like the previously reported biased agonism in GPCRs, for P450 oxidoreductase (POR) and its ligands. The Authors combine a variety of techniques, including computational docking and single molecule FRET, to prove that POR activity and specificity towards CYPs can be biased through modifying POR equilibrium conformational states using small molecules. Targeting such a key node in regulatory or metabolic networks could open many exciting possibilities to modify functioning of metabolic hubs, for example, perturbed by some disease. The novelty of the study is in transferring the concept of GPCRs biased agonism to the idea of biased metabolism, applied here to the metabolic hub of POR and CYPs.

Though I agree that the presented concept is very appealing, I am not sure how realistic it is to regulate the whole network like POR+CYPs with, let's say, one small molecule targeting a highly dynamical enzyme like POR (since design of a drug inhibiting just one enzyme without too many side effects at the cell and organism level is often challenging).

The Authors use a variety of techniques, but they study just the 3 selected compounds, which are claimed (line 90-91) to have "promising effects on the POR function" (this is not a very clear statement to me). Cyclophosphamide is maybe interesting for other reasons, but it has rather mild effect in vitro on both human and sorghum POR (Fig 2B vs. Fig. 3A). Furthermore, the 2 ligands, rifampicin and dhurrin, modulate activity in vitro of POR homologues from human and sorghum in different directions (Fig 2B vs. Fig. 3A). So no common mechanism is identified, and the Authors justify this by sequence differences, which are not analyzed (as pointed out below). The two targets are quite different, different effects of the selected compounds are observed on their activities, which overall seems not so surprising. Finally, the concentrations of these compounds, for which the significant in vitro POR activity change is observed, are quite large (100uM), as the Authors also admit in the summary (l. 404-405). If the concentration is large enough, we may observe the effect of any compound on the target activity, due to non-specific binding.

The title, the abstract and the summary exaggerate a bit the significance of the results. For example, the title says "across kingdom biased (...) metabolism", but the Authors study only the two POR variants from different kingdoms. As the Authors also state, these two POR enzymes share less than 40% of sequence identity and the Authors do not analyze the sequence differences/similarities, as far as I noticed. For a more general outcome, I think, a bioinformatic analysis should be done, if POR sequences are available. The Authors also say in the summary that "each ligand docking on POR stabilizes a distinct equilibrium state that is linked to distinct downstream metabolic outcomes." (l. 406-407). I would say that this is a bit exaggerated. For example, the smFRET results are quite interesting, but I am not sure if the effect of each ligand on the POR activity could be linked to a specific distinct conformational state (when the Results are presented, it is not stated in black and white as in the summary).

The paper could be very interesting if it is backed by more data proving the point. The Authors use the word "correlation" (l. 306: "correlation between conformational sampling and substrate specificity", l. 395: "While our data do not distinguish between the swinging and rotating motion models of POR they provide a correlation between the existence of POR equilibrium conformational states with distinct phenotypic metabolic outcomes.") for few analyzed cases. In my opinion, the study should be further extended to prove the stated hypotheses.

Furthermore, regarding the methodology, I would say the level of detail of computational docking description is not sufficient to repeat the computational experiment (Supplementary methods, docking section). Other comments related to docking simulations:

- l. 154-157: "Cyclophosphamide appeared to cause an increase in POR capacity to reduce Cytc ($149 \pm 31\%$ of control) but had no significant effect in neither the MTT nor RS assay when tested at 10 or 100 μM (Fig 2B). This supports that binding close to FAD and NADPH cofactors does not apriori reduce or eliminate activity." Overall, I think that the conclusions related to docking site of cyclophosphamide vs. its effect on reduction by POR may be too far-fetched. Cyclophosphamide is quite a small ligand, it is additionally quite flexible, and has halogen atoms (possibility of forming halogen bonds). Also, it binds to the very flexible POR. Without further methodological details, I am not able to assess if docking results are convincing.

- As the Authors state, the target is quite dynamic and ligands are docked to only 2 conformers. For example, docking poses in Fig. 1B(de) do not look stable at all - it looks like the ligand is interacting with some flexible loops (short MD following docking simulation may not reliably test the pose stability).

- p.114-117 - Ligands, especially in Site I, dock in the POR centre and the mentioned mutations with disease outcomes are located in the central part of POR. Furthermore, site I, is quite large. So the similar location of disease-causing mutations with binding site location of the ligands may be a coincidence.

Minor comments:

- The usage of word "docking": I am used to hear this word in the context of computational docking; in the paper it is used in place of "binding", which is sometimes misleading.
- l. 89 - "activity of specific CYPs" - unclear statement: what type of activity?
- If I am not missing something, the Authors first describe the experiment with CytC reduction (l. 119-129) and then introduce the same experiment again for 3 other electron acceptors (including CytC).
- l. 199 - typo "extend" > "extent"

Reviewer #3 (Remarks to the Author):

The authors propose that electron transfer functionality by P450 oxidoreductases is modulated by previously uncharacterized ligand binding, and that the differential effects are driven by ligand stabilization of alternative POR conformational states. This has relevance to drug development and metabolic pathway control as tuning of POR activity may allow selective regulation of cytochrome P450s. This mechanism is termed biased metabolism, and is explored through computational docking of ligands to demonstrate binding potential and identify potential binding poses, in vitro and in vivo functional assays measuring POR electron transfer specificity upon treatment with the ligands, and fluorescence microscopy of fluorophore labeled POR to directly measure conformational fluctuations at the single enzyme level. The evidence illustrates the central points that ligand interactions alter POR electron transfer specificity and that ligands induce sampling of distinct POR conformational states, but there are points that could be further clarified.

- 1) A large number of P-value significance tests are performed (Figures 2, 3, 5), were corrections applied for multiple comparisons?
- 2) The term "docking" is used oddly throughout the manuscript (Line 208 docking affinity, 209 docking sites, title, etc.). Typical usage refers to computational modeling of a small molecular binding pose with a protein; however, here it is used for both the computational simulation and interchangeably with "binding".
- 3) Figure 1B is too small to make sense of the ligand binding poses. There is a floating orange line in Fig 1B-A and 1B-B, what is this? Could the positions on the human POR corresponding to where the Sb POR was labeled in the smFRET experiment be highlighted to show their relative positioning to the proposed ligand binding sites (or Figure 4D could have the ligand sites highlighted)?
- 4) Figure 1D and other bar charts, is there a no POR control to confirm that the effect of the ligand is on POR and not on the electron acceptor?
- 5) These docking simulations are not conclusive, statements such as "supports that binding close to FAD and NADPH cofactors does not apriori reduce or eliminate activity" on line 156 is claimed too strongly. Without experimental validation such as crystal structure, the exact binding site is not confirmed and it seems like any ligand that is not exceptionally large would be successfully docked here.
- 6) Can further explanation be provided for how to read Fig 4A traces? Are the y-axis all the same for Fig 4B? How is state occupancy calculated, based on the overlap between gaussians it looks like states are not completely discrete and conformations can be members of multiple states at once?
- 7) The figures and overall writing is well done.

Response Letter

REVIEWER COMMENTS

Reviewer #1 (Remarks to the Author):

The paper entitled "Across kingdom biased CYP-mediated metabolism via small-molecule ligands docking on P450 oxidoreductase" describes: i) how ligands can bind specifically to NADPH cytochrome P450 reductase (POR), ii) how this binding modifies the conformational equilibrium landscape of POR and, iii) in return, how the changes in the conformational equilibrium finally affect differentially electron transfer (ET) from POR to various acceptors. The ultimate claim of the paper is that ET from POR to cytochromes P450 (CYPs) can be biased by ligands.

The conclusions of the paper are **novel and original and extremely important**. The fact that CPR may be biased toward certain CYPs has already been evidenced both in reconstituted systems with variant forms of CPR and from genetic data on CPR polymorphism, leading to the hypotheses that control of specificity by POR can be triggered by either a change in the conformational equilibrium or by mutations that induce redox partner selection. The present

article adds a truly additional and original mechanism for specialization: binding of molecules at the interface between the FMN and FAD domains of POR leads to changes in the conformational equilibrium only, which, in return, potentiates certain redox partners interactions while reducing others.

The discovery of this phenomenon **has strong implications on drug metabolism**: molecules tested here are known to be substrates or inhibitors of human CYPs related to drug metabolism. The results presented here demonstrate that these small compounds induce a modulation of CYPs activities while not being linked to CYPs binding per se, leading to the potential “reuse” of drugs already on the market for other purposes and also potentially explaining adverse drug effects that were not related to CYP direct activation or inhibition.

The claims of the paper are of strong interest to readers outside of the POR/CYP interaction fields because it demonstrates that conformational equilibrium changes within one protein can lead to specific protein-protein interactions and preferential metabolism. On a general point of view, the paper explains the molecular basis for “allosteric metabolic control” by proteins at the centre of metabolic hubs. **This point of view is quite original and gives mechanistic and atomistic explanation of metabolon controls.**

Several points merit attention prior publication.

We are grateful for the reviewers critical and detailed reading of our manuscript, for acknowledging its “strong implications in the field”, and that it is “of strong interest to the readers outside the POR/CYP interaction”. In response to the comment of the referee we have expanded the main text and Supplementary materials to fully address all reviewer’s remarks. Below we have detailed our explicit answer to each of the referee’s comments and outlined the required changes in the manuscript

Docking of POR ligands induce biased specificity towards electron acceptors.

- **Comment 1-1**

Line 89 and further. How these ligands were chosen? In particular, the selection includes molecules that are gold standards for CYP assays. Did the authors have a preliminary screen or did they test all possible drugs? If a preselection was performed, was it done with docking or functional activity assays? Did the authors use a threshold for MW before (potential) screening of the ligands? A small comment on the library that was screened would suffice but it would be quite interesting to know the proportion of ligands that bind POR (docking) within a library of small molecules which have been described to affect CYP activities and, amongst the ones that dock, which proportion is effectively active on POR. This is of particular interest as these molecules are also substrates and/or inhibitors of CYPs themselves. Finally, the use of dhurrin as a test molecule for mammalian POR is surprising. Of course, some of the authors are specialists of the dhurrin pathway for which metabolic control has been demonstrated, but what was the rationale to test it with human POR? Would dhurrin be a general regulator of all POR?

Answer #1-1:

This is a great question and we thank for giving us the chance to rectify it here and in the revised manuscript, as it indeed deserved better description.

We downloaded the annotated subset of chemicals on 49.794 compounds from Chemical Entities of Biological Interest ChEBI (v. 194) and the approved drug subset on 2.355 compounds from Drugbank (v. 15.2). Only compounds containing C,H,N,O,P,S,F,Cl,Br,I were included. After Lipinski Rule-of-Five descriptors were calculated and compounds with a molecular weight above 1.000 removed, the datasets contained 37.708 and 2.035 compounds, respectively. The two datasets were combined and a Principal Component (PC) analysis performed based on the four Lipinski Rule-of-Five descriptors, which were normalized prior to the PC analysis. Preparation of the datasets, calculation of the Lipinski Rule-of-Five descriptors and the PC analysis were all performed via a KNIME workflow. The 8 compounds were selected based on the PC analysis as structurally diverse and based on experimental and/or clinical data that could be assimilated with aberrant CYP function.

We agree that small molecule ligands like rifampicin, cyclophosphamide, ritonavir and warfarin are known to interact with liver CYP2B6 and CYP3A4, and they may dock on additional protein. Abiraterone is also known to interact with CYP17A1 and is indeed used as positive control in the cell assays. As a matter of fact, we never meant to imply that the tested molecules explicitly bind on POR and we clearly stated this both in the cell assays paragraph and in the discussion “Although these ligands may also bind on additional proteins and “..an **extra** layer of regulatory cue”. **The main manuscript results are focused on molecules that we have no evidence of interacting with CYP17, CYP19 or CYP21.** Our combined *in vitro* binding and functional assays, and single molecule studies that are recorded in the absence of any CYP unambiguously affirm that these molecules **can, and do, bind on POR**. This is further compounded by a) our new 1 microsecond MD simulations displaying stable binding of all ligands as well as b) intrinsic fluorescence quenching showing ligand binding close to tryptophan’s, as predicted by MD simulations (see our detailed answer to comment 2-2, 2-3, 2-6 and 2-7 of reviewer 2) and c) Importantly, the assays using MTT the readout of which is not reliant on a CYP function.

The reviewer also found surprising the selection of dhurrin as a ligand, as it is not employed as a pharmaceutical. Dhurrin was chosen as a representative of the family of bioactive plant natural products termed cyanogenic glycosides. More than 3000 plant species have been shown to contain cyanogenic glycosides. This class of bioactive natural products are present in multiple components of the daily diet of humans and their presence are five-times as common in domesticated crop species compared with plants found in natural ecosystems^{1,2} and as such it is important to include a representative of the class from an evolutionary perspective. As components of our foods, they are, in a similar fashion as drugs, chemicals that may interfere with the POR-CYP system.

Would our data provide an answer to the reviewers last question “if dhurrin could be a regulator of all PORs?” The acquired data provide unambiguous evidence that dhurrin a) binds to both human and plant POR and b) affect their electron relaying activity. This is interesting, as human and plant POR only share 38% primary sequence identity, and could indicate this to be a general feature of additional PORs. This is supported by the new docking studies, extended to 1 microsecond MD simulations and sequence alignment that show the amino acids interacting with dhurrin to be conserved across plant, rat and human POR. While such a universal mechanism is very exciting and plausible, its verification would require additional experiments that we reserve to do in the nearest future. We have added an extra sentence on this in the discussion based on reviewer comment.

Changes in the manuscript

To fully address all comments of the reviewer we have:

- a) Added in L90: “Initial ligand selection was based on combination of the annotated subset of chemicals and approved drug subset from Drugbank following Lipinski Rule-of-Five descriptors and principal component analysis (see Supplementary material and Supplementary Fig 1A)”.
- b) Added the detailed description of ligand selection in supplementary information and in Supplementary Fig 1A the PCA analysis for the compound selection.
- c) Highlighted in the main text that “The 3 tested compounds are not known to interact directly with CYP17, CYP19 or CYP21”, in L386, and included in the supplementary methods a section “Selection of chemical compounds for initial studies” describing the above two paragraphs.
- d) Explicitly written twice in the text that rifampicin is known to interact with liver CYPs like CYP2B6 and CYP3A4; in i) L143 The fact that rifampicin and cyclophosphamide are known to interact with liver CYPs like CYP2B6³ and CYP3A4⁴ furthermore shows that direct docking on POR should be taken into consideration when screening for drug-CYP interactions.” and ii) discussion of cell studies in L386 “The 3 tested are not known to interact directly with CYP17, CYP19 or CYP21, albeit rifampicin and cyclophosphamide are known to interact with liver CYPs like CYP2B6 and CYP3A4”.
- e) Explained that abiraterone was used as positive control for CYP17A1 inhibition in cell studies, in L392.
- f) Added in discussion in L247 “Deciphering whether dhurrin is a generic regulator biasing POR-mediated CYP metabolism across all PORs is an exciting possibility that requires additional experimental verification.”

- Comment 1-2

Line 92 and further. In some cases, molecules were not tested at the second concentration (100 μ M). Why? Maybe the authors should explain this in the figure caption. Why did the authors report the SEM instead of the SD? Even if the calculation of the SD from the SEM is quite straightforward, reporting the dispersion of the data points is much more informative than the accuracy of the mean. This is especially true with cyclophosphamide (in Figure 2D) for which the SEM is greater than 1/3rd of the mean value and therefore the SD is greater than 50% of the mean value. This question holds true for the rest of the paper evidently except for Fig. S5 where SD are actually reported.

Answer #1-2

We thank the reviewer for noticing the missing info and apparent inconsistency on some of the graphs, all of which are rectified in the revised version. We note that all three molecules that showed response in all assays and are central for the manuscript, were tested in bulk and in cells at both 10 μ M and 100 μ M. The reason for some apparent inconsistencies were:

- a) Indeed, Mitomycin C and Warfarin were tested only at 10 μ M during our initial screenings. Warfarin showed no measurable effect and was thus not studied any further. Mitomycin C was found to inhibit POR capacity to reduce RS (54 \pm 6 % of control), however, previous studies report Mitomycin C to be directly reduced by POR acting as a substrate, why it was abandoned^{5,6}.
- b) Similarly, the smFRET assays were recorded at slightly different ligand concentrations than 100 μ M so as to optimize the response based on IC50 values; Rifampicin showed sufficient response at 100 μ M, but cyclophosphamide was tested at 200 μ M due to higher IC50. Similarly, dhurrin was tested at 1mM as in this regime it a) elicits a strong measurable effect on POR function b) the response of the CytC and resazurin electron acceptors are different (see also response 1-9).
- c) The missing control spectra of MTT at 100 μ M ligands has been added in Supplementary Fig 7. In the same figure dhurrin and cyclophosphamide control at 200 μ M sufficiently show no effect on fluorescent properties.

- d) Abiraterone control for cell assays was performed at 10 μM as we have shown in the past that it has K_i in nanomolar range 7.

Regarding the error bars and data dispersion. We would like to emphasize strongly that **all conclusions** in the original manuscript **are not based on the error bars** (SEM or SD), **but on Welch unequal variance t-test and calculations of p-values**. Therefore, displaying SEM or SD in the figures does not change any of the conclusions of the manuscript. We have in the revised version replaced all figures to now display both SD error bars but, importantly, all data points on the graphs.

Post-Scriptum

Prompted by the comment 1 of reviewer 3 to apply corrections for multiple comparisons, we performed ANOVA test analysis to decide whether the effect of one or more ligands were significantly different from the control (confidence level: $p < 0.05$) (see detailed response in comment 3-1). In case of a statistically significant difference, Tukey's HSD test was used for pairwise comparison between each of the ligands and the control to determine which of the ligands had a significant effect while correcting for multiple comparisons. The level of significance as determined by Tukey's HSD test is marked by asterisk symbols on the corresponding bar charts. The newly and more accurate calculated p values are in agreement with the initial submission values though the number of stars changed in a couple of cases. Level of significance is marked by asterisk symbols (* $p < 0.05$; ** $p < 0.01$; *** $p < 0.005$). We emphasize that none of the main conclusions of the manuscript is altered based on the new p values.

Changes in the manuscript

To fully address the comment of the referee and abide to Nature publication policy in the revised version we have:

- Clarified in the legend of Supplementary Fig 1 that there were no further studies on warfarin and mitomycin as they did not show promising results.
- Added the extra data set at 50 μM for rifampicin at Supplementary Fig 10.
- Explained L392 that control experiment with the known CYP inhibitor abiraterone were performed at saturating 10 μM concentration and provide a reference showing saturation at nM.
- We have also added in Supplementary Fig 7 the missing control spectra of MTT at 100 μM ligands. In the same figure dhurrin and cyclophosphamide control at 200 μM sufficiently show no effect on fluorescent properties.
- Discussed in all main figure legends that t-test analysis correcting for multiple comparisons was attained by ANOVA test and Tukey's HSD test for pairwise comparison of significance.
- Displayed all data points of replicates on bar plots and furthermore added all data, biological and technical repetitions comprising all main figures in the new Supplementary Tables 2-3 and called this in all relevant places in the main text of the revised manuscript.
- Displayed all error bars as SD instead of SEM.

- Comment 1-3

Line 98 and further. The extended conformation is from rat POR. As stated in the Supp Fig. 2, Sites I-V on human POR do not completely align with Sites I-V on rat POR. Why the authors did not use an extended conformation for human POR, as they did for SbPOR? It would be more logical to analyze the binding on the compact or extended conformations with the same sets of amino acids, even if the identity between human and rat POR is 94%. This could even help get better precision on how the binding affects or is affected by the transition from the compact to the extended form.

Answer #1-3:

This is a great comment and thanks for allowing us to comment on this, here and in the manuscript. To date there is no available crystal structure of the extended conformation of human POR. The only available structure is that of the human-yeast chimera, that exists in a fully extended conformation. It is less accurate to perform docking studies on a chimeric structure as it does not necessarily resemble a native conformation of human POR. We therefore chose to perform the docking studies on the crystal structure from rat as it shows 94% sequence identity. Interestingly, the sequence alignment provided in the revised version combined with the new extended 1 microsecond MD simulations on human POR and new binding studies on both human and plant POR strongly support ligands to bind to evolutionary conserved amino acids (see detailed response to comment (2-2, 2-3, 2-6 of reviewer 2)).

Changes in the manuscript

- We rephrased the 2nd paragraph of "Binding of POR ligands induce biased specificity towards electron acceptors" and explained in the main text (L103-105) that "rat POR shows 94% sequence identity with human POR (see Supplementary Fig 2 for sequence alignment) and was chosen in the absence of a human POR extended conformation and in preference of the available human-yeast chimeric structure."
- Added Supplementary Fig 2 with sequence alignment of human, rat and sorghum POR and
- Discussed the sequence alignment in main text in L503-506.

- **Comment 1-4**

Line 126 and further. "All compounds were tested at 100 μ M using 40 μ M Cytc and 100 μ M NADPH as substrates". In the SI, the authors stated that: "Cytc and RS concentrations were both close to K_m of the given electron acceptor, while MTT and NADPH were in excess amounts". Testing at the K_m value is not necessarily a good idea as it is difficult to sort the effect of the ligands on the saturation behaviour or the k_{cat} . **A full saturation curve of Cytc activity with the four different molecules would be more informative, especially if either the K_m or the k_{cat} or both are modified.** In Figure 2, in the Cytc assay: what was the concentration of CPR used in the assay (approximately?). The relative absorbance change is rather small in 100 seconds: approximately 0.001 OD in 1 min, yielding approximately 50 nM of reduced Cytc per minute for Cyclophosphamide. This tends to indicate that POR was present at very low concentration in the test, between 50 to 100 μ M, if not lower. At these concentrations, POR loses FMN very fast if in the oxidized form prior to NADPH addition. The potential loss of FMN is evidently critical to accurately measure POR activity. In the referenced papers (SI, ref 24-27) the buffer conditions for POR activity tests are different (Ref 24 is using 50 mM KPO4 buffer, Ref 25 is using 50 mM Tris and various concentrations of NaCl, Ref 26 is using 100 mM KPO4 buffer and Ref 27 50 mM Tris). It would be more helpful to precisely describe the conditions of the tests, in particular the salt concentration, buffer used and concentration of POR.

Answer #1-4: We thank the reviewer for the valuable comment, which we have fully addressed by extra experiments on human and plant POR. We performed a full saturation curve for dhurrin on human POR in microsomes (see Supplementary Fig 8 in the revised manuscript). The data are in agreement with Fig 2 of the main text and therefore further augment the conclusions of the manuscript. Since we had already recorded the IC50 binding curves for plant *Sb*POR2b in the original submission, we performed the full Michaelis-Menten kinetics of *Sb*POR2b on liposomes for all 3 ligands at 100 μ M and compared them to POR without ligands (see Supplementary Fig 11 in the revised manuscript). Our data support ligand binding to alter primarily the V_{max} value, supporting biased ligand binding to operate non- or uncompetitively with substrate, confirming our docking studies and MD simulations.

The reviewer also commented on artefacts from potential loss the FMN cofactor, under these experimental conditions. This is indeed a potentially crucial element and we have quantified in the past the FMN loss of POR and furthermore how membrane and lipid environment affect this⁸. However, this is not a potential source of error here: Firstly, we used 10-100 nanomolar concentration of POR in all *in vitro* assays. Specifically, the concentration of hPOR in the assay displayed in Fig 2 is 130 nM (~1 μ g/well). Secondly, all assays reported here are **comparative** and recorded in parallel. So, in the unlikely event of a minute loss of FMN within the experimental time scale, this would be identical across ligands. The observed differences between the 3 ligands and the free POR therefore should correspond to activity variations. This is seconded by our microsomal studies, cell studies and studies at higher concentration reporting output variation of POR.

Changes in the manuscript

- a) Discussed in main text L223 that Michaelis-Menten kinetics of *Sb*POR in the presence of the 3 ligands show ligand to primarily affect V_{max} supporting not competitive agonism. Data are displayed in Supplementary Fig S11.
- b) Added in main text the dhurrin dose-response data for human POR in microsomes (L181 and L252), displayed them in Supplementary Fig 8 and described the methodology in Supplementary Methods.
- c) We have specified in the supplementary information the experimental conditions used for all POR *in vitro* activity assays and discussed explicitly which buffer conditions were used in sections "*In vitro* POR activity assays" and "hPOR dose response activity assays in microsomes".
- d) Added that comparative assays exclude artefacts from potential minute FMN loss in the main text (L173: "Being comparative these assays exclude artefacts that may originate from potential FMN loss").

POR ligand binding and biased specificity pertain across kingdoms.

- **Comment 1-5**

Line 183 and further. The authors present the full curves (IC50) for inhibition and activation. This data is much nicer than just the end points presented in the previous section. **Why the authors did not use it with human POR.** Only IC50 curves can evidence mixed-type behaviours such as dhurrin with Cytc or RS.

Answer #1-5: The IC50 binding curves in the original submission had been recorded for plant sorghum POR. Following the advice of the reviewer we have performed the full binding curve for the most potent ligand, dhurrin, also for human POR in microsomes (see Supplementary Fig 8). In agreement with the pure biophysical system on human POR in proteoliposomes, we observe minute response by MTT, while a more pronounced effect with Cytc and resazurin assay. In the tested concentration regime, we do not observe pronounced mixed type of behaviors as shown for plant POR. Interestingly, direct comparison of human POR data in liposomes with microsomes (Fig 2 and Supplementary Fig 8)

show that while the effects are consistent, their amplitude appears to vary between POR in liposomes and microsomes. This is consistent with the plant POR data in liposomes and detergents and indicates that POR protein and membrane microenvironment can finetune the effect of ligands. (see our detailed answer and literature by us and other in this matter in answer to comment 1-7).

Changes in the manuscript

- a) Highlighted in the main text the full binding curve for dhurrin on human POR in microsomes in L181 and L252 and added in Supplementary Fig 8 the data and Supplementary methods the experimental details.
- b) Explicitly discussed that membrane properties may finetune the effect of ligands L250-252 and also in discussion L485 that ligand effect "...appears to be finetuned by variations in protein and membrane microenvironment."

- Comment 1-6

Line 203 and further. The sentence "The observed effect is inverted at concentrations above 100-500 μ M dependent on the assay, indicating a negative feedback loop type of mechanism downregulating dhurrin production in plants at high dhurrin concentrations" is a bit misleading as the authors describe the effects on alternate electron acceptors. The data presented throughout the article tend to demonstrate that the stimulating or inhibiting effects of the molecules tested are electron acceptor dependent. Therefore, the authors cannot ascertain that the dhurrin effect observed on Cyt_c and RS is identical with CYPs participating in dhurrin synthesis in sorghum (although this is a very good hypothesis that could be easily tested with microsomal fractions).

Indeed, as the reviewer correctly noticed the negative feedback loop is an exciting hypothesis, that to be fully justified may warrant additional experimentation. We had simply outlined this possibility and explicitly written that it "**may indicate...**" and "**can** be studied in the future". The proposal was made because dhurrin has been observed to accumulate at such high (~100 mM) concentrations in sorghum⁹ thus making activation of a feedback loop at these concentration levels a likely event. Dhurrin appears to be binding with similar affinities to the two binding sites of POR (see supplementary Figs 3-5). It is not unlikely therefore that at high concentrations binding to the second site might be achieved which could potentially lead to differential responses. Testing this exciting possibility with microsomal fraction and additional studies is indeed a great experimental idea. We think, however, that deciphering the existence of a feedback loop falls further away from the main scope of this manuscript and might confuse the reader. We will implement this in our coming studies and manuscripts.

Changes in the manuscript

To tone down the argument and address the comment of the reviewer we implemented the above text in the main text L243-246. We also explicitly wrote in L 246-248 that "deciphering whether dhurrin is a generic regulator biasing POR-mediated CYP metabolism across all PORs is an exciting possibility that requires additional experimental verification."

- Comment 1-7

The sentence "Although the amplitude and effects are different in detergent compared to liposomes, it highlights that the observed effects are not induced by altered properties of the lipid bilayer" seems counterintuitive. On the contrary, the observed differences can demonstrate that part of the effects may be attributed to altered properties of the lipid bilayer. Otherwise the effects, if they have absolutely no dependency on the immediate physicochemical environment of POR, should be identical unless the authors think that POR ET properties (as well as dynamics etc.) are different in proteoliposomes vs. detergent. This point should be discussed more extensively, in particular the potential role of the microenvironment of POR (as this is also mentioned further in the paragraph).

Answer #1-7:

We thank the reviewer for allowing us to comment further on this, as it was not clear in the original submission. We fully agree with the referees comment that "... POR ET properties are different in proteoliposomes vs detergent ". As a matter of fact, we recently compared POR behavior in detergents and nanodiscs with that of POR with truncated membrane spanning helix, and reported radically different behavior^{8,10}. Interestingly, membrane properties and topology appears as a common feature underlying the function and spatiotemporal localization of a spectrum of membrane spanning proteins like transporters¹¹⁻¹³ or membrane associated proteins^{14,15}. We have strongly emphasized this in multiple works in Science and Nature journals as well as other multidisciplinary journals¹⁶⁻²¹. Consistent with these, our new data presenting full binding curves of dhurrin on human POR in microsomes (see answer 1-3 above and Supplementary Fig 8) also show minute response variations when compared to POR in liposomes (Fig 2 in the manuscript). The data are qualitatively the same, but dhurrin reduces POR activity in resazurin assay to $82 \pm 6\%$ of control for liposomes, while to $65 \pm 3\%$ for POR in microsomes. Ligands response indeed appears to be finetuned by the membrane and protein environment asserting "POR to respond in a pluripotent way" by regulatory inputs.

Changes in the manuscript

While we are strong advocates of how the lipid properties, but also geometrical or collective membrane features, regulate all aspects of protein function, we believe that delving deeper in this matter here, will confuse non-specialized readers and most certainly dilute unnecessarily the main message of the manuscript. We will be publishing such results in future manuscripts.

- a) We therefore have added an entire new section in L250 that starts with “*Similar finetuning of the effects of ligands by membrane and protein microenvironment was found for hPOR in liposomes and microsomes (see also Supplementary Fig 8 for data on hPOR in microsomes)*”. The section continues with the description above and ends with “*POR protein and membrane microenvironment may thus finetune the effect of ligands and should be taken into consideration when analyzing POR dynamics and function*” (L255-257).
- b) We also added in discussion that biased metabolism “... appears to be finetuned by variations in protein and membrane microenvironment” in L485.

Direct observation of POR conformational sampling and its remodeling by ligands

- **Comment 1-8**

Line 276 and further. The authors should comment on the difference between the calculated distance using FRET in the compact form (44 Å) and the one obtained by modelling (31 Å): 13 Å seems quite a lot, especially for the compact form. Are the dyes positions fixed?

Answer #1-8:

The reviewer asks practically two questions here, a) the interdye distance calculation and b) its correlation to POR states.

- a) Interdye distance calibration is an interesting topic and indeed we, and the single molecule FRET society, have spent considerable effort to ensure proper calibration of FRET values to distance^{22,23}. We integrated this calibration in our recent software to automatically analyze single molecule FRET data using deep learning²⁴. The dyes are attached via flexible linkers to POR so as to ensure free rotation, as it is essential for proper distance calculation. The average dye rotation (shown as spheres in Fig 4D) and distances of 31 Å, 59 Å and 97 Å, are calculated by Monte Carlo simulation (see Fig 4D and Supplementary information) for the compact, intermediate and fully extended homology models of SbPOR2b based on PDB files 3QE2, 3ES9 and 3FJO, respectively. We had explicitly discussed this in our Supplementary Methods Section “Homology modelling and Monte Carlo simulation of dye-dye distance”.
- b) The data on the other hand displayed on the table in Fig 4C correspond to the *average distances of the conformational states S1 to S5, which reflect an equilibrium between multiple conformations*. The calculated distances 76 Å, 64 Å, 57 Å, 48 Å and 44 Å as such correspond to the *average distance between two or more interconverting conformations* and do not necessarily align with interdye distances extracted from static homology models. We had explicitly discussed this in the 2nd paragraph of smFRET studies (L294-L323 of the revised version) and provided evidence for the equilibrium in Supplementary Fig 13.

Changes in the manuscript

To fully address the comment of reviewer and ease the general reader to apprehend this concept we did the following:

- a) To clarify the equilibrium point more clearly we wrote in the figure legend “Each FRET state may reflect an equilibrium between multiple conformations”.
- b) We wrote in Fig 4D legend that the conformations correspond to the compact, intermediate and extended conformations.
- c) Wrote in 4D and supplementary materials that distance calculation were based on Monte Carlo simulation and added this in main text.
- d) We now also wrote in the main text (L304-L307) that data analysis was done based on a machine learning framework that allowed smFRET classification and FRET to distance calibration, and in the supplementary information that “the 14.0 Å long linker length ensures free rotation of the dyes which is essential for distance calibration”.
- e) We explicitly added in main text L306 “see Supplementary Fig 13 and Supplementary methods for FRET to distance calibration”.

- **Comment 1-9**

Line 291 and further. “To evaluate the effect of ligands on POR conformational sampling, we measured smFRET on POR exposed to ligand concentrations well above IC50”. The rifampicin concentration used (100µM) is effectively well above the EC50. However, for cyclophosphamide, 200 µM is not at all “well above” EC50 (159 and 228 µM). For dhurrin, it would have been very informative to add another concentration, the one where SbPOR activated RS reduction (100 µM of dhurrin) to analyse it in view of the effect at 1 mM where SbPOR reduces RS reduction.

Answer #1-9: This is indeed an inaccurate phrasing and we thank the referee for noticing it. We have explicitly added in the main text the missing concentrations and added that for cyclophosphamide we used higher concentration due to the different IC50. The dhurrin concentration of 1mM at the original submission was selected at a regime that potentially

maximizes output. This was based on the facts of a concentration where dhurrin a) elicits a strong measurable effect on POR function b) the response of the CytC and resazurin electron acceptors are different.

The reviewer also suggested that it might be more informative to add an additional concentration at the regime of 100 or 10 μ M for the single molecule FRET studies. As we wrote in comment (2-6 of reviewer 2, the manuscript's data unambiguously support a correlation of conformational sampling to biased metabolism, but not direct assessment of conformational equilibrium states to metabolic outcomes. Performing this extra set of experiments could provide some extra hints of the concentration dependence of biased conformational sampling. However, it will neither reveal **how** biased conformational sampling relates to function nor **which** of the conformational equilibrium states is assigned to each of the distinct metabolic outcomes, the reviewer eludes to.

Given these experiments would provide little added value to the main claims of the manuscript, we decided not to perform the additional and time-consuming single molecule FRET experiments consequently unnecessarily delaying further publication. We intend to do so in the nearest future and follow up work.

Changes in the manuscript

- a) Added that rifampicin was at 100 μ M in L344 and that "cyclophosphamide was tested at 200 μ M due to higher IC50 values (see Fig 3)" in L347 of the main text.
- b) Added in main text that "Dhurrin was tested at 1 mM that appear to result in opposite responses for resazurin and CytC reduction and was found to induce a small..." in L350.

POR ligands bias steroid hormone metabolism in human cells and microsomes

Comment 1-10

Line 327 and further. "Rifampicin caused no significant effect on CYP17A1 activity, but reduced CYP21A2 activity at 100 μ M (73 \pm 2% of control). Considering the SD and not the SEM, the decrease of activity of CYP17A1 with rifampicin at the two concentrations and not be so different from the effect at 100 μ M rifampicin for CYP21A2.

Answer #1-10-:

All the effects described as significant were reliant on Welch's t-test analysis which is now changed to ANOVA test and Tukey's HSD test in the revised manuscript to correct for multiple comparisons between ligands/experiments. We note, however, that none of the conclusions have changes as a result of the new t-tests. **Only data with p-values less than 0.05 are described as significant.** If p-values are below 0.05 we reject the null hypothesis that the two samples have equal means. We have detailed this in response 1-2 of this reviewer.

Changes in the manuscript

To clarify this and fully address the comment 1-2 of the reviewer we have added all data points on top of the bar charts, changed error bars to display SD instead of SEM in the revised manuscript and provided Supplementary Tables 2-3 outlining number of replicates and SD for each experiment. See also answer to comment 1-2 and response 3-1 to reviewer 3.

- Comment 1-11

"Cyclophosphamide enhanced both CYP17A1 hydroxylase and lyase activity at 100 μ M (111 \pm 1% and 132 \pm 5% of control, respectively), while having a small yet statistically insignificant effect on CYP21A2 activity (113 \pm 5% of control)". I believe the authors wanted to write significant instead of insignificant? In any case, if the 111 \pm 1% is considered significant for the authors, 113 \pm 5% is certainly significant as well!

Answer #1-11:

Please see our response to comment 1-10 above and 1-2 of this reviewer. Please see also answer to reviewer 3, comment 3-1.

- Comment 1-12

Overall, in this part, one control is missing: the absence of direct interactions of rifampicin, cyclophosphamide and dhurrin with CYP17A1, CYP21A2 and CYP19A1. Rifampicin and cyclophosphamide are known to interact with several drug-related CYPs. The probability that they interact with the three above mentioned CYPs is not small and should be assessed, directly on microsomal fractions, by spectral changes (Type I or Type II). For dhurrin, the control should be performed as well.

Answer #1-12:

We have partially answered this in comment 1-1 and we expanded and specified our response further here. We agree that small molecule ligands like rifampicin, cyclophosphamide, ritonavir, amiodarone and warfarin interact with *liver* CYP2B6 and CYP3A4, and they may dock on additional proteins. As a matter of fact, we never meant to imply that the tested molecules explicitly bind to POR and we clearly stated this both in the cell assays paragraph and in the discussion "Although these ligands may also bind on additional proteins and "...an **extra** layer of regulatory cue". To the best of

our knowledge, however, there is no evidence of direct docking of the three tested ligands on any of the CYP17A1, CYP21A2 and CYP19A1.

The main manuscript results are focused on molecules that we have no evidence of interacting with CYP17A1, CYP19A1 or CYP21A2.

Our combined *in vitro* binding and functional assays, single molecule and *in vivo* data clearly support that these molecules **can, and do, bind on POR**. This is further compounded by a) our new 1 microsecond MD simulations displaying stable binding of all ligands as well as b) intrinsic fluorescence quenching showing ligand binding close to tryptophans and tyrosines, as predicted by MD simulations (see our detailed answer to comment (2-6 and 2-7 of reviewer 2) and c) Importantly, the assays using MTT the readout of which is not reliant on CYP function.

Changes in the manuscript

Acknowledging that this might still be unclear in the manuscript we have:

Explicitly written twice in the text that rifampicin is known to interact with liver CYPs like CYP2B6 and CYP3A4; in

- a) L143: "The observation that rifampicin and cyclophosphamide are known to interact with liver CYPs like CYP2B6³ and CYP3A4⁴ furthermore shows that direct docking on POR should be taken into consideration when screening for drug-CYP interactions".
- b) discussion of cell studies in L386: "The 3 tested are not known to interact directly with CYP17, CYP19 or CYP21, albeit rifampicin and cyclophosphamide are known to interact with liver CYPs like CYP2B6 and CYP3A4".
- c) added in L397: "Because MTT assay is not reliant on CYPs function, these data further support that POR docking can facilitate biased CYP activation in cellular environment".
- d) added in discussion L483 that "Biased metabolism would operate in parallel with existing and well-studied regulatory cues controlling CYP mediated metabolism".

Minor points

Main text:

- Line 49: the generalization is misleading. Metabolic control does not only apply to redox protein partners and cytochromes P450, the latter only represent part of the metabolism.
- Corrected it now reads "Dynamic assemblies and function of multiple redox protein partners with cytochromes P450 (CYPs) and NADPH-dependent cytochrome P450 oxidoreductase (POR) orchestrate control of multiple metabolic cascades across kingdoms.
- Line 67: its instead of Its (after the colon).
Corrected
- Line 69: the reference 19 describes how mutations in the FAD domain can switch specificity of POR from NADPH to NADH and, as a consequence, induce a differential coupling with human CYPs. Other articles that also describe specificity changes due to mutations on POR should be cited to be completely exhaustive: Esteves, F. et al. The Role of the FMN-Domain of Human Cytochrome P450 Oxidoreductase in Its Promiscuous Interactions With Structurally Diverse Redox Partners. *Front Pharmacol* 11, 299 (2020). Campelo, D. et al. Probing the Role of the Hinge Segment of Cytochrome P450 Oxidoreductase in the Interaction with Cytochrome P450. *Int J Mol Sci* 19, (2018).
- Corrected. We thank the reviewer for the comment and have cited the two articles in the revised manuscript.
- Line 127: comma between activity and support not necessary. Corrected.
- Line 241: The SAXS reference number 6 is incomplete: the reference number 9 and another reference (Huang, W.-C., Ellis, J., Moody, P. C. E., Raven, E. L. & Roberts, G. C. K. Redox-Linked Domain Movements in the Catalytic Cycle of Cytochrome P450 Reductase. *Structure* 21, 1581–1589 (2013)) should be added to be completely exhaustive.
Corrected.

Supplementary Information:

- Line 290: 3ES9 instead of 3ES.
- Corrected.
- There is a general problem with the reference numbering in the SI. For example, reference 2 (line 37) in the SI is probably wrong; line 70 refs 24-27 possibly wrong as well (not all of them); line 81 ref. 42; line 92 ref. 42; line 101 ref. 42; line 112 ref. 42; line 133 ref. 42; line 141 ref. 47 (which does not exist in the reference section); line 148 ref. 40; line 156 and 159, ref. 48 and 49 are not present in the reference section, etc. Please recheck all references in the SI.

- We thank the reviewer for noting the misalignment between two documents. This has been corrected in the revised version.

Reviewer #2 (Remarks to the Author):

The Authors propose a concept of biased metabolism, like the previously reported biased agonism in GPCRs, for P450 oxidoreductase (POR) and its ligands. The Authors combine a variety of techniques, including computational docking and single molecule FRET, to prove that POR activity and specificity towards CYPs can be biased through modifying POR equilibrium conformational states using small molecules. **Targeting such a key node in regulatory or metabolic networks could open many exciting possibilities to modify functioning of metabolic hubs, for example, perturbed by some disease.** The novelty of the study is in **transferring the concept of GPCRs biased agonism to the idea of biased metabolism, applied here to the metabolic hub of POR and CYPs.**

Though I agree that the presented concept is very appealing, I am not sure how realistic it is to regulate the whole network like POR+CYPs with, let's say, one small molecule targeting a highly dynamical enzyme like POR (since design of a drug inhibiting just one enzyme without too many side effects at the cell and organism level is often challenging).

We thank the reviewer for acknowledging the novelty of the work and its potential to "open many exciting possibilities to modify functioning metabolic hubs and diseased states". Fully addressing the reviewers concerns by multiple extra experiments further improved the quality of the manuscript.

Prior to fully and in detail addressing all the specific points, we wish to highlight that we **did not state that one small molecule targeting POR is the sole regulator of the entire network of POR-P450 reactions.** On the contrary, we do acknowledge - and all authors have extensive work supporting - the existence of multiple layers of regulatory cues controlling metabolism^{19,25,26}. Our data firstly showcase that a) small molecules **can, and do, dock on POR**, thus reveal POR as a potential pharmacological target and b) **small molecules docking on POR may act as an "additional cue"** controlling POR-CYPs metabolism. This work's very observation that at least one intermediate of the dhurrin metabolism docks on POR and biases its output on CYPs, further compounds that the overall observed biased metabolism may be the convolution of multiple interactions with small molecules. Some of the small molecules may only be present at specific developmental stages or in specific tissues, considerably expanding the regulatory control layers. We had explicitly discussed this in the manuscript (see L494 "biased metabolism represents an extra, hitherto unrecognized, layer of regulation capable of controlling"). Acknowledging it might still be misinterpreted, we have now added in the discussion that "Biased metabolism may operate in parallel with existing, and well-studied, regulatory cues controlling CYP-mediated metabolism" in the discussion in L483.

Below we have detailed our answers and extra experimental work that the reviewer requested (5 completely new sets of experiments) that include: 3 new dose response curves for dhurrin on hPOR, extended 1 microsecond molecular dynamics (MD) simulations of the 5 docking poses on the closed form of hPOR, binding studies based on intrinsic fluorescence quenching confirming ligand docking on both human and plant POR close to tryptophans as predicted by MD simulations, and lastly Michaelis-Menten kinetics on plant POR in the presence of each of the 3 ligands. Besides this we have performed a sequence alignment between the human, rat and plant POR isoforms and discussed in the manuscript. Lastly, we tuned down the tone in all places the reviewer found us excessive. We hope they have addressed in full all reviewers concerns and will prompt the reviewer to recommend publication in *Nature Communications*.

- Comment 2-1

The Authors use a variety of techniques, but they study just the 3 selected compounds, which are claimed (line 90-91) to have "promising effects on the POR function" (this is not a very clear statement to me). Cyclophosphamide is maybe interesting for other reasons, but it has rather mild effect in vitro on both human and sorghum POR (Fig 2B vs. Fig. 3A).

We thank the reviewer for critically reading the manuscript and noticing the missing clarification, which is also noticed by reviewer 1. As we detailed in response #1-1 to reviewer 1, we downloaded the annotated subset of chemicals on 49.794 compounds as well as the approved drug subset on 2.355 compounds from Drugbank (v. 15.2). Only compounds containing C,H,N,O,P,S,F,Cl,Br,I were included. After Lipinski Rule-of-Five descriptors were calculated and compounds with a molecular weight above 1.000 removed, the datasets contained 37.708 and 2.035 compounds, respectively. The two datasets were combined and a Principal Component (PC) analysis performed based on the four Lipinski Rule-of-Five descriptors, which were normalized prior to the PC analysis. Preparation of the datasets, calculation of the Lipinski Rule-of-Five descriptors and the PC analysis were all performed via a KNIME workflow. The 8 compounds were selected based on the PC analysis as structurally diverse and based on experimental and/or clinical data that could be assimilated with aberrant CYP function. Out of these compounds, the 3 compounds with higher

response (see Fig 1D) were progressed for further studies. Indeed, cyclophosphamide response was lower as compared to dhurrin and rifampicin, albeit it was higher than amiodarone, cimetidine, mitomycin, ritonavir and warfarin, that were not tested further.

We highlight that the *in vitro* assays of main Figures 2 and 3 are a *means to an end* showcasing in a clean biophysical system POR mediated biased specificity by these ligands. The cell studies showcase cyclophosphamide to indeed pertain to biased specificity towards the biological and pharmacological relevant function of CYP17A1.

Changes in the manuscript

To rectify the lack of clarity we have performed the following changes in the revised manuscript:

- a) Added in L90 "Initial ligand selection was based on combination of the annotated subset of chemicals and approved drug subset from Drugbank following Lipinski Rule-of-Five descriptors and principal component analysis (see Supplementary material and Supplementary Fig 1A)".
- b) Added the detailed description of ligand selection in Supplementary information and in Supplementary Fig 1A the principal component analysis for the compound selection.

- Comment 2-2

Furthermore, the 2 ligands, rifampicin and dhurrin, modulate activity *in vitro* of POR homologues from human and sorghum in different directions (Fig 2B vs. Fig. 3A). So no common mechanism is identified, and the Authors justify this by sequence differences, which are not analyzed (as pointed out below). The two targets are quite different, different effects of the selected compounds are observed on their activities, which overall seems not so surprising.

Answer #2-2:

We fully agree our data show that the same ligand may induce different response on the two different POR isoforms, or that different ligands on the same POR may elicit diverse responses. We had highlighted that in the discussion (now in L229-235 and L501 and onwards). We respectfully, yet strongly, disagree however that this undermines a common mechanism. As a matter of fact, **our data are completely in line with the biased agonism of GPCRs where the actual downstream response is dependent on the specific ligand-protein interactions:** This can be manifested either by structurally diverse ligands, or even enantiomeric ligands, **eliciting diverse downstream signaling response**^{27,28} or the same ligand binding on two isoforms of a receptor resulting in diverse downstream signaling effects, described as isoform bias^{29,30}.

Congruently, our data display the same ligands a) *bind* on both POR isoforms, b) bind at similar docking sites and c) structurally diverse ligands can bind at the same site but interact with different amino acids, differentially biasing POR conformational dynamics and allosterically controlling specificity towards CYPs. They thus strongly support a **common underlying regulatory mechanism** that operates via ligand binding on POR, **the effect of which**, varies depending on the fine-tuned, specific allosteric interactions induced by ligand binding.

This is further compounded by the following two series of completely new experiments we provided based on comment 2-7 of this reviewer and shortly discussed below:

- a) Extended 1 microsecond MD simulations of the three tested ligands on the closed form of human POR showing stable binding and dominant interactions with amino acids that are primarily conserved across human, rat and plant POR isoforms as shown by sequence alignment (see Supplementary Fig 2+5 and Supplementary Movies 1-3).
- b) Intrinsic fluorescence quenching experiments on both human and plant POR isoforms confirming binding of the tested ligands close to conserved aromatic residues (see also comment 2-3 of this reviewer).

These studies confirm common binding sites Ia and Ib and dominant ligand interactions with amino acid residues that are conserved across POR isoforms (e.g. W679 and Y458). They also reveal that the ligands interact with multiple amino acids in different ways (Supplementary Fig 5) supporting differential allosteric responses.

Changes in the manuscript

We acknowledge this was not clear in the manuscript, so we have added:

- a) In the main text L229: "Interestingly, human and plant isoforms respond differently to the same ligand, in line with differential response of GPCRs isoforms by the same ligand^{29,30}".
- b) In the main text L118: "1 microsecond molecular dynamics (MD) simulations confirmed these interactions and revealed stable binding for all three ligands (see Supplementary Fig 5 and Supplementary Movies 1-3). Tested ligands were found to interact primarily with amino acids that are conserved between human, rat and plant POR (see Supplementary Fig 2 for sequence alignment), albeit each ligand interacted in different ways with additional specific amino acids (see Supplementary Fig 5)".
- c) In the main text discussion L487: "...variations on ligand protein interactions, originating either by structurally diverse ligands or by different POR isoforms, may be manifested as varied metabolic responses."

- Comment 2-3

Finally, the concentrations of these compounds, for which the significant *in vitro* POR activity change is observed, are quite large (100 μ M), as the Authors also admit in the summary (l. 404-405). If the concentration is large enough, we may observe the effect of any compound on the target activity, due to non-specific binding.

Answer #2-3:

We fully agree with the reviewer that the measured affinities are low for any medicinal application at this stage, as we explicitly discussed this in the summary stating that the ligands serve as proof of concept (L477), the working concentrations are rather high (L479) and our findings pave the way for the *in silico* design of pathway-specific biased ligands (L500). We emphasize however, that not all tested compounds displayed an effect at 100 μ M (see amiodarone, cimetidine and ritonavir in Supplementary Fig 1B) suggesting that the reported interactions are to some extent specific. This is further substantiated by the new insights we have gained with the series of completely new experiments:

- a) The long 1 microsecond MD simulations on human POR fully affirm stable binding to POR for all three ligands and display dominant interactions with amino acid residues that are primarily conserved across human, rat and plant POR isoforms (see also response to comment 2-2 and 2-7).
- b) Intrinsic fluorescence quenching studies on both human and plant POR isoforms confirm binding of the tested ligands close to conserved, aromatic residues as predicted by MD simulations (see also response to comment 22 and 2-7).

Combined, these insights ascertain that the ligands **bind specifically and stably to POR**, albeit with low affinity. Furthermore, the plant metabolite dhurrin is known to accumulate at very high (~100 mM) concentrations in sorghum ρ thus making binding to POR a likely event at these concentrations.

Changes in the manuscript

To fully and in detail address the comment of the reviewer we have:

- a) Added the MD simulations for all three ligands on human POR in Supplementary Fig 5, Supplementary Movies 1-3, described it in the main text (L118) and added the experimental details in supplementary methods.
- b) Discussed in main text the intrinsic fluorescence quenching studies (L130) explicitly stating "The recorded quenching was in agreement with MD simulations showing strong interaction of all three ligands with tryptophan and tyrosines (W679, Y458, Y481) in both site Ia and Ib", added a relevant section in the supplementary material and displayed the data in Supplementary Fig 6.

- Comment 2-4

The title, the abstract and the summary exaggerate a bit the significance of the results. For example, the title says "across kingdom biased (...) metabolism", but the Authors study only the two POR variants from different kingdoms. As the Authors also state, these two POR enzymes share less than 40% of sequence identity and the Authors do not analyze the sequence differences/similarities, as far as I noticed. For a more general outcome, I think, a bioinformatic analysis should be done, if POR sequences are available.

Answer #2-4:

We have changed the title and section heading as the referee pointed out that only two POR variants from different kingdoms are studied. Along the same line, the new title starts with "Biased cytochrome P450-mediated metabolism" emphasizing the concept of biased metabolism rather than the "across kingdom" in the original submission. Following the comment of the referee, we have performed sequence alignment of the relevant human, rat and plant POR isoforms and included this in Supplementary Fig 2 (see also response to comment 2-2, 2-3 and 2-7).

Changes in the manuscript

- a) Title is now "Biased cytochrome P450-mediated metabolism across mammal and plant kingdoms via small-molecule ligands docking on P450 oxidoreductase."
- b) Section title "POR ligand binding and biased specificity pertains across *mammal and plant* kingdoms".
- c) Added sequence alignment in Supplementary Fig 2 confirming that amino acids interacting with ligands in MD simulations and docking studies are evolutionary conserved.
- d) Discussion of sequence alignment in the main text (L120 and L503) and added that "additional, across kingdoms, bioinformatics analyses are required to investigate whether hotspots for ligand binding pertain across the entire spectrum of POR isoform" (L506).

- Comment 2-5

The Authors also say in the summary that "each ligand docking on POR stabilizes a distinct equilibrium state that is linked to distinct downstream metabolic outcomes." (l. 406-407). I would say that this is a bit exaggerated. For example, the smFRET results are quite interesting, but I am not sure if the effect of each ligand on the POR activity could be linked to a specific distinct conformational state (when the Results are presented, it is not stated in black and white as in the summary).

Answer #2-5

We have rectified this in the revision version by:

Changes in the manuscript

- a) Revised this sentence highlighting that "...binding on POR redistributes the conformational equilibrium, consequently altering interaction with CYPs and downstream metabolism. Future studies may fully confirm each equilibrium state to be directly linked to distinct downstream metabolic outcome."
- b) Added an extra figure (Fig 6) displaying a hypothetical energy landscape where conformational equilibrium states are re-distributed by ligands, a mechanistic trait that results in biased metabolic cascades. Note that more than one energy minimum is redistributed in alignment with the comment of the reviewer.

- Comment 2-6

The paper could be very interesting if it is backed by more data proving the point. The Authors use the word "correlation" (l. 306: "correlation between conformational sampling and substrate specificity", l. 395: "While our data do not distinguish between the swinging and rotating motion models of POR they provide a correlation between the existence of POR equilibrium conformational states with distinct phenotypic metabolic outcomes.") for few analyzed cases. In my opinion, the study should be further extended to prove the stated hypotheses.

Answer #2-6:

We agree with the reviewer that more data proving a direct link between binding of each ligand to specific conformational states and distinct metabolic outcomes would always be more interesting. We wish to emphasize that it required almost half a century of intense studies for the GPCR community to transit from the first data of biased ligands³¹ (also named functional selectivity, ligand directed signaling or biased agonism) to actual direct evidence of ligand mediated stabilization of distinct conformations, and their links to distinct signaling outcomes. This resulted in a wealth of separate high impact publications by the leading authorities (e.g. Nobel laureates Kobilka and Lefkowitz) often spanning decades of work. Their work included computational studies³², FRET³³, NMR³⁴, crystallographic evidence^{35,36} and cell studies³⁷ to mention a few.

Our work harnesses and combines the insight of a spectrum of the above techniques, all of which provide strong evidence of POR-mediated biased metabolism. Concurring with the reviewer's comment that the "*presented concept is very appealing*", "*could open many exciting possibilities to modify functioning of metabolic hubs (...) perturbed by some disease*", may indeed merit further justification of the concept. In the response to reviewer comments, we have performed **5 additional series of experimental and computational studies**:

- a) 1 microsecond MD simulations for all 5 poses of the 3 ligands on human POR (see also answer 1-1 for reviewer 1 and 2-2, 2-3 and 2-7 for this reviewer) revealing stable binding of ligands and interactions with specific amino acid residues of POR (Supplementary Fig 5).
- b) Intrinsic fluorescence quenching based binding assays for both human and sorghum POR. They both confirm the binding of the 3 ligands to be stable and furthermore so close to aromatic residues of POR (see Supplementary Fig 6). The presence of tryptophan and tyrosine (W679 and Y481) close to the binding site Ia, furthermore supports that our docking studies and MD simulations correspond to the actual ligand binding locations on POR.
- c) Sequence alignment for human, rat and plant sorghum POR revealing the 3 tested ligands to primarily interact with amino acids that are conserved across human, plant and rat species signifying their importance for function and indicating the existence of hotspots for ligand binding and modulation of function.
- d) Michaelis-Menten studies of plant POR for Cytc reduction in the presence of each of the tested ligands dhurrin, cyclophosphamide and rifampicin (see also answer to comment 1-4 for reviewer 1 and Supplementary Fig 11). The large changes in Vmax support that biased ligand binding is not competitive to substrate.
- e) Dose-response studies of dhurrin on human POR in microsomes for all three substrates (Cytc, MTT and RS, see Supplementary Fig 8). Our data on microsomes are in agreement with the data for human POR on liposomes (Fig 2) asserting the manuscript conclusions.

We are happily reporting that all above studies conclusively demonstrate the tested ligands bind stably to POR, alter POR substrate specificity and function. They thus both affirm our assertions of POR-mediated biased metabolism in cells and the potential implication of these findings for controlling metabolism.

Changes in the manuscript

- a) We have added in the main text that stable binding to POR for all 3 ligands was further supported by 1 microsecond molecular dynamics simulations and added a section in methods and Supplementary Fig S5 showing key data from the MD simulations.
- b) We have added in the main text that we confirmed the docking studies and MD simulations with intrinsic fluorescence quenching (L130-134), added the methodology in Supplementary methods and the results in Supplementary Fig 6.
- c) Added in several places that extended 1 microsecond MD simulations confirmed binding to POR.

- d) Added in main text the discussion of new dhurrin dose-response studies (L250-252) on human POR in microsomes followed by Supplementary Fig 8 and experimental details in Supplementary methods.
- e) Discussed in the main text (L223) the Michaelis-Menten kinetics of plant POR and added a new Supplementary Fig 11 displaying the results.

Comment 2-7

a) Furthermore, regarding the methodology, I would say the level of detail of computational docking description is not sufficient to repeat the computational experiment (Supplementary methods, docking section). Other comments related to docking simulations:

b) - l. 154-157: "Cyclophosphamide appeared to cause an increase in POR capacity to reduce Cytc (149±31% of control) but had no significant effect in neither the MTT nor RS assay when tested at 10 or 100 µM (Fig 2B). This supports that binding close to FAD and NADPH cofactors does not apriori reduce or eliminate activity." Overall, I think that the conclusions related to docking site of cyclophosphamide vs. its effect on reduction by POR may be too far-fetched.

c) Cyclophosphamide is quite a small ligand, it is additionally quite flexible, and has halogen atoms (possibility of forming halogen bonds). Also, it binds to the very flexible POR. Without further methodological details, I am not able to assess if docking results are convincing.

- As the Authors state, the target is quite dynamic and ligands are docked to only 2 conformers. For example, docking poses in Fig. 1B(de) do not look stable at all - it looks like the ligand is interacting with some flexible loops (short MD following docking simulation may not reliably test the pose stability).

d)- p.114-117 - Ligands, especially in Site I, dock in the POR centre and the mentioned mutations with disease outcomes are located in the central part of POR. Furthermore, site I, is quite large. So the similar location of disease-causing mutations with binding site location of the ligands may be a coincidence.

Answer #2-7:

We took very seriously the comment of the reviewer on the "insufficient description", "non-convincing docking studies" and that "... short MD may not reliably test the pose stability". We have detailed our answers based on our new acquired data below.

Answer 2-7 a)

Thanks for noticing the insufficient "...computational docking description" we have elaborated in supplementary methods by adding 2 paragraphs with detailed description of the docking studies and MD simulations.

Answer 2-7b)

The reviewer insightfully noticed a section that could have been described better. A central element of POR function is the conformational changes from the open/extended state to the closed state bringing FAD close to FMN and facilitating electron transfer between them. The FMN subsequently relays the electrons to CYPs reducing the central iron of the heme from the ferric to the ferrous state as required for oxygen binding. Ligand binding between, or close to, the FAD and FMN may thus hinder this motion completely eliminating activity of POR. The fact that cyclophosphamide enhances activity towards Cytc supports that the binding in this docking site does not generically eliminate POR function by nonspecifically hindering POR motions prohibiting electron transfer between FAD and FMN.

Answer 2-7c) The referee is concerned that cyclophosphamide is a small and flexible molecule, and the docking poses does not seem to make any specific interactions with the protein and, accordingly, may not represent stable binding modes.

To fully address the concern of the reviewer on stable binding, we have extended the originally reported 10 nanosecond MD simulations with new 1 microsecond simulations and all the docking poses are essentially stable (see also answers to comments 2-2 and 2-3 and see all binding energies of the initial docking in Supplementary Table 1). The RMSD values for the five MD simulations are presented in a new Supplementary Fig 5, and Supplementary Movies 1-3. The average RMSD values for the complexes in Fig 1B(de), corresponding to cyclophosphamide and dhurrin in Site Ib, are 2.0 ± 0.2 Å and 2.1 ± 0.3 Å, respectively, indicating that the docking poses are stable. Thus, despite cyclophosphamide being a small ligand, it appears that binding is stable. Careful inspection of the new Supplementary Fig 5 reveals a dominant interaction of the cyclophosphamide with W679. If this represents the correct ligand binding mode, we should expect a quenching in the intrinsic fluorescence of POR. Our completely new intrinsic fluorescence quenching studies did indeed show quenching of POR fully confirming the validity of our docking studies, MD simulations and thus the claims of this work.

Answer 2-7d) This is indeed a valid concern and we thank the reviewer for pointing this out. The docking and extended MD simulations provide some answers and improved our understanding of these interactions. In Supplementary Fig 5 and Supplementary Table 1 of the revised manuscript we provide all amino acids interacting with the ligands. Cyclophosphamide and dhurrin appear to interact directly and shortly with G539 and rifampicin with R600. Interestingly,

the mutation of these residues result in aberrant CYP activation and metabolic disorders. Fully exploring the disease-causing POR mutants with ligands presents an exciting possibility of alleviating existing metabolic disorders but requires model building of the mutated proteins, docking of ligands, subsequent MD simulations, bulk and single molecule assays that are rather lengthy processes beyond this manuscript, but certainly in our future work.

Changes in the manuscript

- a) Added in main text (L118) that docking simulations of 1 microsecond confirmed stable ligand binding,
- b) Detailed in Supplementary methods how the docking studies and MD simulations were performed and added a new Supplementary Fig 5 and Supplementary Movies 1-3.
- c) Discussed in the main text in L177 that “This supports that binding close to FAD and NADPH cofactors does not reduce or eliminate activity by non-specifically hindering POR motions and prohibiting electron transfer between FAD and FMN”.
- d) Discussed in main text (L130-134) that ligand binding was confirmed by intrinsic fluorescence quenching studies.
- e) Detailed in supplementary methods how intrinsic fluorescence quenching studies were performed and added in Supplementary Fig 6 responses based on tryptophan quenching by all 3 ligands on both human and plant POR isoforms.
- f) Highlighted further in Fig 1 legend that “*See Supplementary Figs 2 and 3 higher magnification and detailed interactions and Supplementary Table 1 for binding energies*”.
- g) Altered the main text in L120-125: “Tested ligands were found to interact primarily with amino acids that are conserved between human, rat and plant POR (see Supplementary Fig 2 for sequence alignment), albeit each ligand interacted in different ways with additional specific amino acids (see Supplementary Fig 5). Interestingly, all three ligands were found to interact directly with either one or both of amino acid residues G539 and R600 of human POR (see Supplementary Fig 4 and Supplementary Table 1). Mutations of the very same residues are associated with POR deficiency”. We also elaborated further on this in the discussion L496-499 “Targeting POR may serve as a way of controlling POR-CYP interactions and regulate CYP-mediated metabolic pathways. *This is further compounded by the direct interaction of the tested ligands with amino acids where mutation is associated with POR deficiency and specifically disorder of sexual development and production of sex steroids.*”
- h) Added in main text L130-134 and in section “POR ligand binding and biased specificity pertain across mammal and plant kingdoms” in L213 that we performed binding assays based on intrinsic fluorescence quenching for both human and plant POR and they display binding for all three ligands.

- **Minor comments:**

- The usage of word “docking”: I am used to hear this word in the context of computational docking; in the paper it is used in place of “binding”, which is sometimes misleading.

We have corrected this throughout the paper and replaced “docking” to “binding” except when referring to the computational process.

- l. 89 - “activity of specific CYPs” - unclear statement: what type of activity?

We have replaced with function so as to encompass both activity and expression.

- If I am not missing something, the Authors first describe the experiment with CytC reduction (l. 119-129) and then introduce the same experiment again for 3 other electron acceptors (including CytC).

We have added in the figure legend of Fig 2 that “Data for CytC from Fig 1 are included for comparison”.

- l. 199 - typo “extend” > “extent”

We have corrected this in the revised version

Minor further changes

Note, prompted by comment 1-2 of reviewer 1 and comment 3-1 of reviewer 3, we have changed all error bars to display SD in the revised version and performed ANOVA and Tukey’s HSD tests to correct p-values for multiple comparisons, why **all** figures have been replaced. We emphasize that none of the conclusions are altered by the new p-values and in some cases the levels of significance have improved.

Reviewer #3 (Remarks to the Author):

The authors propose that electron transfer functionality by P450 oxidoreductases is modulated by previously uncharacterized ligand binding, and that the differential effects are driven by ligand stabilization of alternative POR conformational states. This has relevance to drug development and metabolic pathway control as tuning of POR activity may allow selective regulation of cytochrome P450s. This mechanism is termed biased metabolism, and is explored through computational docking of ligands to demonstrate binding potential and identify potential binding poses, *in vitro* and *in vivo* functional assays measuring POR electron transfer specificity upon treatment with the ligands, and fluorescence microscopy of fluorophore labeled POR to directly measure conformational fluctuations at the single enzyme level. The evidence illustrates the central points that ligand interactions alter POR electron transfer specificity and that ligands induce sampling of distinct POR conformational states, but there are points that could be further clarified.

We thank the reviewer for critically reading the manuscript, accepting it for publication acknowledging its relevance “to drug development and metabolic pathway control” and her/his valuable comments. Fully and in detail addressing all of them helped us further improve the quality of the manuscript.

- **Comment #3-1**

A large number of P-value significance tests are performed (Figures 2, 3, 5), were corrections applied for multiple comparisons?

Answer #3-1: We had used Welch's t-test to compare the control distribution to each of the ligand distributions for each experiment. Welch t-test was chosen since the control and ligand sample sizes were different in some cases. The reviewer is correct that t-test for multiple comparisons is more suited for this manuscript. **We have rectified this in the revised version.** We emphasize that, as expected, **none of the main conclusion of the manuscript are altered.** T-tests correcting for multiple comparisons were now performed to calculate p-values. For each set of experiments, an ANOVA test was performed to decide whether the effect of one or more ligands were significantly different from the control (confidence level: $p < 0.05$). In case of a statistically significant difference, Tukey's HSD test was used for pairwise comparison of each experimental condition to determine which of the ligands differed significantly from the control. The level of significance as calculated by Tukey's HSD test is marked by asterisk symbols (* $p < 0.05$; ** $p < 0.01$; *** $p < 0.005$).

Changes in the manuscript

- We have added the above description in the supplementary methods (section “Statistical analysis of *in vitro* activity data”)
- We have replaced error bars to report SD instead of SEM in **all** figures and plotted individual data points on top of bar charts in Fig 1-3, Fig 5 as well as Supplementary Fig 1B to further display sample sizes and uncertainties.
- The level of significance is corrected for multiple comparisons based on ANOVA and Tukey's HSD tests which is explicitly stated in all figure legends.
- We have added Supplementary Tables 2-3 detailing all information used for statistical analysis of the *in vitro* and cell assays. This includes the mean and standard deviation of each experiment as well as number of biological and technical replicates. We called this in the main text in all sections discussing data analysis.
- We have explicitly added “significant” to denote all responses with significant p values ($p < 0.5$) are observed, both in main text and figure legends.

- **Comment #3-2**

The term “docking” is used oddly throughout the manuscript (Line 208 docking affinity, 209 docking sites, title, etc.). Typical usage refers to computational modeling of a small molecular binding pose with a protein; however, here it is used for both the computational simulation and interchangeably with “binding”.

Answer #2: Thanks for noticing the inconsistency. We have corrected this throughout the manuscript and replaced “docking” to “binding” except when referring to the computational process.

- **Comment #3-3**

Figure 1B is too small to make sense of the ligand binding poses. There is a floating orange line in Fig 1B-A and 1B-B, what is this? Could the positions on the human POR corresponding to where the Sb POR was labeled in the smFRET experiment be highlighted to show their relative positioning to the proposed ligand binding sites (or Figure 4D could have the ligand sites highlighted)?

Answer #3:

We agree that the small figures in Figure 1B showing the protein as a cartoon may not offer a complete characterization of the shape of the binding site. This is in the interest of space. Detailed illustration of the shape of the binding sites,

the actual binding modes and the protein-ligand contacts are shown in Supplementary Fig 3-4 and Supplementary Table 1.

The cartoon representations of the proteins are z-clipped in order not to obscure the view of the ligands. The floating orange lines in Figure 1B-A and 1B-B are artefacts of this process. We have removed the lines to improve clarity.

Changes in the manuscript

- 1) Removed floating orange lines in 1B-a and 1B-b.
- 2) Highlighted in Fig 1 legend that “See Supplementary Figs 2 and 3 higher magnification and detailed interactions and Supplementary Table 1 for binding energies”.

- Comment #3-4

Figure 1D and other bar charts, is there a no POR control to confirm that the effect of the ligand is on POR and not on the electron acceptor?

Answer #3-4:

This is indeed a very valuable comment and we have taken great care to correct the missing information. All data in Fig 1D and the rest of the graphs were baseline corrected using “no POR” controls which in all cases were negligible as compared to the DMSO controls in the presence of POR (the slopes were practically zero). Furthermore, all data were corrected for potential photophysical effects imposed by ligands. Errors were propagated across all steps. We refer to Supplementary Fig 7 of the revised submission for spectral controls and calibration, and the “in vitro POR activity assays” section in the supplementary methods for further details.

Changes in the manuscript

Acknowledging that the above may have not been fully clear, we have:

- a) Added in figure legend of Fig 2: “The bar plot represents the mean \pm SD of at least three measurements normalized to DMSO controls with propagated error (see Supplementary methods for details) (...) All data are corrected for potential ligand photophysical effect (see Supplementary Fig. 7)”.
- b) Explicitly added in Supplementary methods section “*In vitro* POR activity assays” that all data were background corrected and controls without POR showed negligible activity.

- Comment #3-5

These docking simulations are not conclusive, statements such as “supports that binding close to FAD and NADPH cofactors does not *a priori* reduce or eliminate activity” on line 156 is claimed too strongly. Without experimental validation such as crystal structure, the exact binding site is not confirmed and it seems like any ligand that is not exceptionally large would be successfully docked here.

Answer #3-5:

The comment of the referee is a crucial point of the manuscript and we took it very seriously during the revision process. Crystallographic evidence or cryoEM, as we did recently for CRISPR²³, is a process that takes months to years. Accordingly, we performed the next best studies a) extended MD simulation of 1 microsecond and b) binding assays based on intrinsic fluorescence quenching (see also response 2-6 and 2-7 to reviewer 2).

The originally reported 10 nanosecond MD simulations are now extended to 1 microsecond simulations that fully confirm all the docking poses to be essentially stable. The RMSD values for the five MD simulations are presented in a new Supplementary Fig 5 and Supplementary Movies 1-3. The average RMSD values for the complexes in Fig 1B(de), corresponding to cyclophosphamide and dhurrin in Site Ib, are 2.0 ± 0.2 Å and 2.1 ± 0.3 Å, respectively, indicating that the docking poses are stable. Although cyclophosphamide is a small ligand, it appears to be very stable in the binding mode identified by computational docking.

Careful inspection of the MD simulations in new Supplementary Fig 5 reveals an interaction of cyclophosphamide and dhurrin with W679 and Y458 but also rifampicin with Y481. If therefore the MD simulations are correct, we should expect a quenching in the intrinsic fluorescence of POR. We therefore employed intrinsic fluorescence quenching assays to further confirm independently the binding of ligand on both human and sorghum POR. Consistent with docking studies and MD simulation, the intrinsic fluorescence quenching studies show strong quenching of both human and plant POR by all three ligands confirming that **ligand binding occurs in position as predicted by the MD simulation**, fully supporting the validity of our docking studies and MD simulations and thus the claims of this work.

Changes in the manuscript

In light of the new experimental data sets and to fully address the comment of the reviewer, we:

- a) Added in main text (L118) that docking simulations of 1 microsecond confirmed stable ligand binding,
- b) Detailed in Supplementary methods how the docking studies and MD simulations were performed, added Supplementary Movies 1-3 and Supplementary Fig 5 displaying the MD simulation data for all 5 poses of the 3 ligands.

- c) Discussed in main text the intrinsic fluorescence quenching studies (L130-134) based on proximity to tryptophan and tyrosines, added the data in Supplementary Fig 6 and a relevant section in the supplementary material describing the experimental details.
- d) Added in section “POR ligand binding and biased specificity pertain across mammal and plant kingdoms” in L213 that we performed binding assays based on intrinsic fluorescent quenching for both human and plant POR and they confirm binding of all three ligands.

- **Comment #3-6**

Can further explanation be provided for how to read Fig 4A traces? Are the y-axis all the same for Fig 4B? How is state occupancy calculated, based on the overlap between gaussians it looks like states are not completely discrete and conformations can be members of multiple states at once?

Answer #3-6:

The methodology we used here is called Alternating-Laser Excitation (ALEX) single molecule FRET^{22,38}. The method operates by rapidly switching between donor and acceptor excitations while recording in parallel donor and acceptor emissions. In doing so, it ensures both donor and acceptor are present (not bleached) and allows for proper calibration of signal to distance fluctuations, as we and others have shown^{22-24,39}. Fig 4a displays the time trajectories of FRET data. The top row displays the Donor (green) and Acceptor (red) intensities over time (s) at 532 nm excitation. The middle row displays the acceptor only intensity (red) at 640 nm excitation. The bottom row displays the EFRET value (orange) calculated with calibration factors. The idealized FRET value determined from HMM fitting is displayed in blue.

This information was originally provided in what is now Supplementary Fig 12, however, acknowledging it was not clear in the main text for the interdisciplinary audience of Nature Communications, we have added this information in the figure legend of main Fig 4 in the revised version.

The referee lastly asks how is each state occupancy calculated, as the distributions overlap. We took great care in this analysis, as well as in our previous publications^{23,24}. Briefly, every gaussian is defined by its mean value and its standard deviation (sigma). In the gaussian mixture models of Fig 4 the actual number of Gaussians is defined by Bayesian Information Criterion (BIC) that objectively determines the underlying number of states and thus the correct number of Gaussians, avoiding overfitting. While fitting with increased number of gaussian will result in better fitting the model penalizes addition of extra fitting parameters. The overall distribution is fit using the expectation maximization algorithm that provides the statistically best fit. Because it is based on unbinned likelihood fitting, it is unbiased by potential artefacts arising from data binning. This fit outputs the mean, standard deviation and weight of each gaussian within the gaussian mixture model. Based on this, the actual contribution of each gaussian to the overall distribution is accurately extracted. We have detailed this information in the methods of our recent papers in Cell and eLife^{23,24}. The latter discusses an automated software called DeepFRET, that implements methodology to rapidly analyze single molecule FRET independently of human intervention based on a machine learning framework comprising a deep neural network.

Lastly, we wish to highlight that the data displayed in Fig 4B correspond to the actual distribution of all data points of EFRET values throughout the trajectories in Fig4a. Each FRET state is discrete and its position and lifetime is very accurately extracted by HMM analysis, however, the actual data are more noisy. (See for example the blue trace vs the actual orange data in last right bottom panel of Fig 4a: The HMM finds discrete states (centered at ~0.7 and ~0.85) but each data point (orange line) is distributed around each state). The width of the EFRET distribution depends on noise in data acquisition and analysis, faster protein dynamics, photo physics etc. Because of this, each of the states in panel 4b has a width instead of a single value.

Changes in manuscript

Acknowledging this was not clear in the manuscript we have provided this extra clarification in the main text and SI of the revised version.

- a) We have added the missing Y axis in Fig 4B. It is “Probability density” thus the integral of each distribution is 1. The y axis in fig 4a corresponds to intensity for each of the channels and excitation laser lines.
- b) We have also added extra info of how smFRET analysis and gaussian mixture model fitting was performed in Supplementary section “quantitative image analysis of smFRET data”.
- c) We have added in main text (L304-307) that “rapid and agnostic annotation and classification of the single molecule FRET data was carried out using DeepFRET²⁴, our recently published methodology based on machine learning (see Supplementary methods for details on data fitting, occupancy extraction and FRET to distance calibration)”.

- **Comment #7**

The figures and overall writing is well done.

We thank the reviewer for the nice comment.

Bibliography

1. Jones, D. A. Why are so many food plants cyanogenic? *Phytochemistry* **47**, 155–162 (1998).
2. Gleadow, R. M. & Møller, B. L. Cyanogenic glycosides: synthesis, physiology, and phenotypic plasticity. *Annu. Rev. Plant Biol.* **65**, 155–185 (2014).
3. El-Serafi, I., Afsharian, P., Moshfegh, A., Hassan, M. & Terelius, Y. Cytochrome P450 oxidoreductase influences CYP2B6 activity in cyclophosphamide bioactivation. *PLoS One* **10**, e0141979 (2015).
4. Moon, J.-Y. *et al.* GC-MS-based quantitative signatures of cytochrome P450-mediated steroid oxidation induced by rifampicin. *Ther Drug Monit* **35**, 473–484 (2013).
5. Wang, S.-L., Han, J.-F., He, X.-Y., Wang, X.-R. & Hong, J.-Y. Genetic variation of human cytochrome p450 reductase as a potential biomarker for mitomycin C-induced cytotoxicity. *Drug Metab. Dispos.* **35**, 176–179 (2007).
6. Wang, Y. *et al.* Distinct roles of cytochrome P450 reductase in mitomycin C redox cycling and cytotoxicity. *Mol. Cancer Ther.* **9**, 1852–1863 (2010).
7. Malikova, J. *et al.* CYP17A1 inhibitor abiraterone, an anti-prostate cancer drug, also inhibits the 21-hydroxylase activity of CYP21A2. *J. Steroid Biochem. Mol. Biol.* **174**, 192–200 (2017).
8. Laursen, T. *et al.* Single molecule activity measurements of cytochrome P450 oxidoreductase reveal the existence of two discrete functional states. *ACS Chem. Biol.* **9**, 630–634 (2014).
9. Kojima, M., Poulton, J. E., Thayer, S. S. & Conn, E. E. Tissue Distributions of Dhurrin and of Enzymes Involved in Its Metabolism in Leaves of Sorghum bicolor. *Plant Physiol.* **63**, 1022–1028 (1979).
10. Moses, M. E., Hedegård, P. & Hatzakis, N. S. Quantification of functional dynamics of membrane proteins reconstituted in nanodiscs membranes by single turnover functional readout. *Meth. Enzymol.* **581**, 227–256 (2016).
11. Zoghbi, M. E., Cooper, R. S. & Altenberg, G. A. The Lipid Bilayer Modulates the Structure and Function of an ATP-binding Cassette Exporter. *J. Biol. Chem.* **291**, 4453–4461 (2016).
12. Akyuz, N. *et al.* Transport domain unlocking sets the uptake rate of an aspartate transporter. *Nature* **518**, 68–73 (2015).
13. Li, M. *et al.* Effects of membrane curvature and pH on proton pumping activity of single cytochrome bo₃ enzymes. *Biochim. Biophys. Acta Bioenerg.* **1858**, 763–770 (2017).
14. Bohr, S. S.-R., Thorlaksen, C., Kühnel, R. M., Günther-Pomorski, T. & Hatzakis, N. S. Label-Free Fluorescence Quantification of Hydrolytic Enzyme Activity on Native Substrates Reveals How Lipase Function Depends on Membrane Curvature. *Langmuir* **36**, 6473–6481 (2020).
15. Šrejber, M. *et al.* Membrane-attached mammalian cytochromes P450: An overview of the membrane's effects on structure, drug binding, and interactions with redox partners. *J Inorg Biochem* **183**, 117–136 (2018).
16. Hatzakis, N. S. *et al.* How curved membranes recruit amphipathic helices and protein anchoring motifs. *Nat. Chem. Biol.* **5**, 835–841 (2009).
17. Larsen, J. B. *et al.* Membrane curvature enables N-Ras lipid anchor sorting to liquid-ordered membrane phases. *Nat. Chem. Biol.* **11**, 192–194 (2015).
18. Rosholm, K. R. *et al.* Membrane curvature regulates ligand-specific membrane sorting of GPCRs in living cells. *Nat. Chem. Biol.* **13**, 724–729 (2017).
19. Laursen, T. *et al.* Characterization of a dynamic metabolon producing the defense compound dhurrin in sorghum. *Science* **354**, 890–893 (2016).

20. Li, M. *et al.* Single Enzyme Experiments Reveal a Long-Lifetime Proton Leak State in a Heme-Copper Oxidase. *J. Am. Chem. Soc.* **137**, 16055–16063 (2015).
21. Hatzakis, N. S. *et al.* Single enzyme studies reveal the existence of discrete functional states for monomeric enzymes and how they are “selected” upon allosteric regulation. *J. Am. Chem. Soc.* **134**, 9296–9302 (2012).
22. Hellenkamp, B. *et al.* Precision and accuracy of single-molecule FRET measurements—a multi-laboratory benchmark study. *Nat. Methods* **15**, 669–676 (2018).
23. Stella, S. *et al.* Conformational Activation Promotes CRISPR-Cas12a Catalysis and Resetting of the Endonuclease Activity. *Cell* **175**, 1856–1871.e21 (2018).
24. Thomsen, J. *et al.* DeepFRET, a software for rapid and automated single-molecule FRET data classification using deep learning. *Elife* **9**, (2020).
25. Flück, C. E., Parween, S., Rojas Velazquez, M. N. & Pandey, A. V. Inhibition of placental CYP19A1 activity remains as a valid hypothesis for 46,XX virilization in P450 oxidoreductase deficiency. *Proc. Natl. Acad. Sci. USA* **117**, 14632–14633 (2020).
26. Bonomo, S. *et al.* Promising Tools in Prostate Cancer Research: Selective Non-Steroidal Cytochrome P450 17A1 Inhibitors. *Sci. Rep.* **6**, 29468 (2016).
27. Coudrat, T., Christopoulos, A., Sexton, P. M. & Wootten, D. Structural features embedded in G protein-coupled receptor co-crystal structures are key to their success in virtual screening. *PLoS One* **12**, e0174719 (2017).
28. Sato, T., Kawasaki, T., Mine, S. & Matsumura, H. Functional Role of the C-Terminal Amphipathic Helix 8 of Olfactory Receptors and Other G Protein-Coupled Receptors. *Int. J. Mol. Sci.* **17**, (2016).
29. Marti-Solano, M. *et al.* Combinatorial expression of GPCR isoforms affects signalling and drug responses. *Nature* **587**, 650–656 (2020).
30. Riddy, D. M. *et al.* Isoform-Specific Biased Agonism of Histamine H3 Receptor Agonists. *Mol. Pharmacol.* **91**, 87–99 (2017).
31. Lefkowitz, R. J. & Williams, L. T. Catecholamine binding to the beta-adrenergic receptor. *Proc. Natl. Acad. Sci. USA* **74**, 515–519 (1977).
32. Salminen, T. *et al.* Three-dimensional models of alpha(2A)-adrenergic receptor complexes provide a structural explanation for ligand binding. *J. Biol. Chem.* **274**, 23405–23413 (1999).
33. Granier, S. *et al.* Structure and conformational changes in the C-terminal domain of the beta2-adrenoceptor: insights from fluorescence resonance energy transfer studies. *J. Biol. Chem.* **282**, 13895–13905 (2007).
34. Bokoch, M. P. *et al.* Ligand-specific regulation of the extracellular surface of a G-protein-coupled receptor. *Nature* **463**, 108–112 (2010).
35. Rasmussen, S. G. F. *et al.* Crystal structure of the human beta2 adrenergic G-protein-coupled receptor. *Nature* **450**, 383–387 (2007).
36. Manglik, A. *et al.* Crystal structure of the μ -opioid receptor bound to a morphinan antagonist. *Nature* **485**, 321–326 (2012).
37. Kinzer-Ursem, T. L., Sutton, K. L., Waller, A., Omann, G. M. & Linderman, J. J. Multiple receptor states are required to describe both kinetic binding and activation of neutrophils via N-formyl peptide receptor ligands. *Cell Signal.* **18**, 1732–1747 (2006).
38. Lee, N. K. *et al.* Accurate FRET measurements within single diffusing biomolecules using alternating-laser excitation. *Biophys. J.* **88**, 2939–2953 (2005).
39. Lerner, E. *et al.* Toward dynamic structural biology: Two decades of single-molecule Förster resonance energy transfer. *Science* **359**, (2018).

REVIEWERS' COMMENTS

Reviewer #1 (Remarks to the Author):

The paper entitled "Across kingdom biased CYP-mediated metabolism via small-molecule ligands docking on P450 oxidoreductase" describes: i) how ligands can bind specifically to NADPH cytochrome P450 reductase (POR), ii) how this binding modifies the conformational equilibrium landscape of POR and, iii) in return, how the changes in the conformational equilibrium finally affect differentially electron transfer (ET) from POR to various acceptors. The ultimate claim of the paper is that ET from POR to cytochromes P450 (CYPs) can be biased by ligands. The authors have now considered the points that all three reviewers addressed and have done a great job in modifying the manuscript. The conclusions are even stronger now.

Reviewer #2 (Remarks to the Author):

The Authors propose a concept of biased metabolism, like in GPCRs, to apply to POR and its ligands. They use computational and experimental techniques to prove that POR regulatory activity can be biased by small molecule ligands. As the Authors admit, the whole concept of biased metabolism is transferred from the case of GPCRs. The novelty is its application to POR regulation of CYP activity. The Authors combine a variety of techniques to study this phenomenon, including computational docking and single molecule studies (smFRET). Overall, the hypothesis stated in the title and the abstract sounds really interesting.

The Authors have put a lot of effort to improve the manuscript, all of my critical comments included in the previous review were addressed. However, I have a few additional comments:

It may be hard to observe unbinding events in classical MD on the sub-microsecond timescale.

Especially, ligand residence times may be longer, when the binding site is quite deep/enclosed and both the ligand and binding site are flexible (<http://dx.doi.org/10.1016/j.drudis.2013.02.007>, <http://dx.doi.org/10.1016/j.cbpa.2010.06.176>). Furthermore, looking at the MD movies:

- they are recorded from such a perspective that it is hard to track the details of the ligand dynamics (ligands are located behind protein secondary structure and a cofactor); also, it is hard to get an idea how enclosed is the final MD-refined binding site.

- Movie 1: cyclophosphamid - unbinds from the original binding site at least once,

- Movie 2: dhurrin - the initial binding pose is different than the final one,

- Movie 3: rifampicin changes its binding mode.

Ligand RMSD is overall stable, but it can be seen that for at least the two above compounds that the predicted binding mode has changed significantly. This shows the predicted docking pose conformational instability (though still does not disprove the POR ligand binding hypothesis). What additional data, in my opinion, could be helpful to better understand the ligand conformational dynamics and binding to POR:

- a 2D interaction diagram of the docked and final MD-refined stable ligand pose with the receptor (of a main cluster representative) would be interesting - to see specific interactions,

- it might be informative to see the parameters such as exposure or enclosure Sitemap parameters for the identified binding sites.

Regarding the ligand binding, fluorescence quenching results seem quite convincing to me, though I am not an expert on this technique. The only problem for me here is the lack of a negative control (a POR non-binder in addition to the 3 studied ligands), measured at the same concentration.

Reviewer #3 (Remarks to the Author):

Novel and well-executed work, the additional experiments and revisions have addressed my comments and greatly improved the quality of the manuscript. There are a few remaining typos to be fixed, mainly in the supplemental (Sup fig 5 missing axis labels, Sup fig 8 denisity to density, Sup fig 11 units unclear). The display of the Michaelis-Menten parameters in a table may be more appropriate than a bar chart (Sup fig 11), and the Kcat parameter rather than Vmax should be shown to normalize for protein concentration.

Response Letter

REVIEWERS' COMMENTS

Reviewer #1 (Remarks to the Author):

The paper entitled "Across kingdom biased CYP-mediated metabolism via small-molecule ligands docking on P450 oxidoreductase" describes: i) how ligands can bind specifically to NADPH cytochrome P450 reductase (POR), ii) how this binding modifies the conformational equilibrium landscape of POR and, iii) in return, how the changes in the conformational equilibrium finally affect differentially electron transfer (ET) from POR to various acceptors. The ultimate claim of the paper is that ET from POR to cytochromes P450 (CYPs) can be biased by ligands. The authors have now considered the points that all three reviewers addressed and have done a great job in modifying the manuscript. The conclusions are even stronger now.

ANSWER:

We are thankful to the reviewer for acknowledging the impact of our work and accepting our revisions. We agree that the critical feedback and careful revisions have led to significant improvements of the manuscript and even stronger conclusions.

Reviewer #2 (Remarks to the Author):

The Authors propose a concept of biased metabolism, like in GPCRs, to apply to POR and its ligands. They use computational and experimental techniques to prove that POR regulatory activity can be biased by small molecule ligands. As the Authors admit, the whole concept of biased metabolism is transferred from the case of GPCRs. The novelty is its application to POR regulation of CYP activity. The Authors combine a variety of techniques to study this phenomenon, including computational docking and single molecule studies (smFRET). Overall, the hypothesis stated in the title and the abstract sounds really interesting. The Authors have put a lot of effort to improve the manuscript, all of my critical comments included in the previous review were addressed. However, I have a few additional comments:

It may be hard to observe unbinding events in classical MD on the sub-microsecond timescale. Especially, ligand residence times may be longer, when the binding site is quite deep/enclosed and both the ligand and binding site are flexible

(<http://dx.doi.org/10.1016/j.drudis.2013.02.007>,

<http://dx.doi.org/10.1016/j.cbpa.2010.06.176>).

Furthermore, looking at the MD movies:- they are recorded from such a perspective that it is hard to track the details of the ligand dynamics (ligands are located behind protein secondary structure and a cofactor); also, it is hard to get an idea how enclosed is the final MD-refined binding site

- Movie 1: cyclophosphamid - unbinds from the original binding site at least once,- Movie 2: dhurrin - the initial binding pose is different than the final one,- Movie 3: rifampicin changes its binding mode. Ligand RMSD is overall stable, but it can be seen that for at least the two above compounds that the predicted binding mode has changed significantly. This shows the predicted docking pose conformational instability (though still does not disprove the POR ligand binding hypothesis). What additional data, in my opinion, could be helpful to better understand the ligand conformational dynamics and binding to POR:- a 2D interaction diagram of the docked and final MD-refined stable ligand pose with the receptor (of a main cluster representative) would be interesting - to see specific interactions,- it might be informative to see the parameters such as exposure or enclosure Sitemap parameters for the identified binding sites. Regarding the ligand binding, fluorescence quenching results seem quite convincing to me, though I am not an expert on this technique. The only problem for me here is the lack of a negative control (a POR non-binder in addition to the 3 studied ligands), measured at the same concentration.

ANSWER:

We thank the reviewer for the critical feedback and acknowledge the concerns regarding MD simulations and intrinsic fluorescence quenching. The referee returns to the reversibility of the binding of the ligands on POR. We wish to highlight that **we never meant to imply that binding is irreversible, on the contrary** we provided evidence and clearly stated **that binding is weak and therefore reversible**, provided micromolar IC50s and explicitly discussed that further improving it by ligand design is required. Understanding “... *the ligand conformational dynamics*” is indeed an interesting perspective but beyond the scope of the work. We have however addressed once again the comment of the reviewer below in the point-by-point response below.

MD movies

The movies were recorded from the same view as the figures in the text to facilitate a direct comparison between figures and movies. While we agree with the referee that this perspective may not be the best to illustrate the binding of the ligands, we would prefer to keep it so as to maintain consistency throughout the manuscript. To address the comment, we have provided an additional Supplementary Movie 4 (see below) displaying close-ups of the ligand binding sites.

For all the MD simulations, we observe some changes in the binding mode during the initial part of the simulations, the equilibration phase. This is not surprising, but rather, quite common when the initial protein structure is determined by X-ray crystallography and, accordingly, refined by another force field than used for the MD simulations. Careful inspection of the RMSD values we had displayed in Supplementary figure 5 show that indeed after 100ns the ligands equilibrate in the binding sites displaying overall stable binding as the reviewer notes. While we could have excluded the first 100 ns of our 1000 ns simulations, that result in equilibration and only have displayed the part illustrating that the ligands remain in the binding site, i.e. binding is stable, we choose to include the complete datasets, and furthermore so to indicate this detail in the figure legend.

We have added in main text when describing the MD simulation and in Supplementary figure 5 that “all ligand display stable binding upon an initial equilibration phase”]

To illustrate further that binding is indeed stable we have produced three 12 sec movies (Supplementary Movie 4) with the frames from the simulations after 100, 200, ..., 1000 ns for cyclophosphamide, dhurrin, and rifampicin. For each compound, the first view corresponds to the same perspective as on the figures in the text and the second view is a closeup of the ligand in the binding site. Dhurrin and cyclophosphamide bind deep in the site between the cofactors, whereas rifampicin binds closer to the surface and also displays a high degree of flexibility. All three ligands remain in the binding site and remain relatively stable after equilibration of ~100ns.

Interaction diagrams

We fully agree that specific interactions are important for understanding the docking pose. As a matter of fact, we had provided the most important intermolecular protein-ligand interactions for the docked poses in the figure legends of Supplementary Figures 3 and 4 and listed all contacts within 5 Å from the docked ligands in the bottom part Supplementary Table 1. The intermolecular interactions observed during the MD simulations are summarized as bar diagrams in Supplementary Figure 5. We respectfully, yet strongly disagree, however, that comparing intermolecular interactions for the docked poses with the interactions observed during the MD simulations, would provide any further useful information, as the two calculations represent different situations. The docking is based on a single low-energy conformation of the protein obtained experimentally by X-ray crystallography, whereas the poses sampled during the MD simulations, performed at 300 K where both the protein and ligand are allowed to move freely, do not necessarily represent low-energy conformations. This is further compounded by our data that indeed display RMSD values supporting that ligands equilibrate in the binding sites. Thus, to justify a detailed comparison, we think that poses from the MD simulations should be relaxed (energy minimized) prior to comparison. The intermolecular interactions from the docking are not as important as the interactions observed during the MD simulations, since the latter represents an ensemble of structures interacting without any constraints.

SiteMap parameters

Two extra columns have been added in the top part of Supplementary Table 1 with the exposure and enclosure scores for the binding sites identified by the SiteMap analysis:

- The exposure score for the Site I in the closed form of POR (3QE2) is 0.53, which is only slightly higher than the average of 0.49 for tight-binding sites according to the SiteMap manual. The exposure score for the Site I in the open form of POR (3ES9) is 0.59 suggesting that Site I in the open form is marginally less likely to be characterized as a potential tight-binding site.
- The enclosure score for Site I in the closed and open forms of POR (3QE2 and 3ES9, respectively) are 0.74 and 0.70 indicating, as expected, that the site in the closed form is less exposed to the surrounding solvent and more likely to represent a tight-binding site. In the SiteMap manual, it is stated that the average for tight-binding sites is 0.78.

Overall, the SiteMap analysis supports that Site I and to some extent Site II can serve as potential tight-binding sites as we had already explicitly stated in the manuscript.

Intrinsic fluorescence quenching

To address the comment of the reviewer we have added glucose as a negative control in Supplementary Fig 6 showing no or minimal effect on intrinsic fluorescence at the relevant concentration as expected. The results further support that the tested ligands cyclophosphamide, rifampicin and dhurrin bind specifically to binding sites with aromatic residues as close contacts in agreement with our MD simulations (interactions with conserved residues W679 and Y458).

Reviewer #3 (Remarks to the Author):

Novel and well-executed work, the additional experiments and revisions have addressed my comments and greatly improved the quality of the manuscript. There are a few remaining typos to be fixed, mainly in the supplemental (Sup fig 5 missing axis labels, Sup fig 8 density to density, Sup fig 11 units unclear). The display of the Michaelis-Menten parameters in a table may be more appropriate than a bar chart (Sup fig 11), and the Kcat parameter rather than Vmax should be shown to normalize for protein concentration.

ANSWER:

We thank the reviewer for acknowledging the novelty and implications of our work. The remaining typos pointed out by the reviewer, have all been fixed in the revised version:

- Supplementary Fig 5, panel B: Axis labels are added on both x- and y-axis.
- Supplementary Fig 11: Units are clarified and Michaelis-Menten parameters displayed in a table instead of bar chart.
- Supplementary Fig 13, panel B: Y-axis label typos are fixed ("density" to "density").